# Endophilin A2 regulates B-cell endocytosis and is required for germinal center and humoral responses

Dessislava Malinova[1,2,*] iD, Laabiah Wasim[1], Rebecca Newman[1], Ana Martínez-Riaño[1],
Niklas Engels[3] iD & Pavel Tolar[1,4,**] iD

## Abstract

Antigen-specific B-cell responses require endosomal trafficking to regulate antigen uptake and presentation to helper T cells, and to control expression and signaling of immune receptors. However, the molecular composition of B-cell endosomal trafficking pathways and their specific roles in B-cell responses have not been systematically investigated. Here, we report high-throughput identification of genes regulating B-cell receptor (BCR)-mediated antigen internalization using genome-wide functional screens. We show that antigen internalization depends both on constitutive, clathrin-mediated endocytosis and on antigen-induced, clathrin-independent endocytosis mediated by endophilin A2. Although endophilin A2-mediated endocytosis is dispensable for antigen presentation, it is selectively required for metabolic support of B-cell proliferation, in part through regulation of iron uptake. Consequently, endophilin A2-deficient mice show defects in GC B-cell responses and production of high-affinity IgG. The requirement for endophilin A2 highlights a unique importance of clathrin-independent intracellular trafficking in GC B-cell clonal expansion and antibody responses.

**Keywords** antigen uptake; B-cell responses; endocytosis; endophilin A2; germinal center

**Subject Categories** Immunology; Membranes & Trafficking

## Introduction

Robust antibody responses are crucially dependent on the B cells' ability to internalize, process, and present foreign antigens. Antigens are recognized by the B-cell receptor (BCR), and their binding triggers rapid BCR endocytosis and trafficking into lysosomal antigen-processing compartments, delivering antigenic peptides for loading onto MHC class II molecules (MHCII) (Lanzavecchia, 1985; Amigorena et al, 1994). Display of peptide-loaded MHCII on B-cell surface solicits interaction with helper T cells, which, together with antigen-induced BCR signaling, stimulates metabolic and transcriptional B-cell activation that culminates in formation of germinal centers (GCs) and production of high-affinity IgG antibodies. Thus, BCR-mediated antigen internalization critically determines the outcome of B-cell responses by controlling the number and repertoire of peptide-loaded MHCII molecules available for recognition by cognate T cells. Furthermore, endocytic trafficking of the BCR modulates BCR signaling strength and duration, in the rapidly proliferating activated B cells (Hoogeboom & Tolar, 2016). Consequently, B-cell responses are sensitive to mutations or inhibitors impairing the endolysosomal system (Onabajo et al, 2008; Chaturvedi et al, 2011; Chatterjee et al, 2012; Veselits et al, 2014). However, the molecular composition of B-cell endocytic and intracellular trafficking pathways remains poorly understood.

The best characterized mechanism of BCR internalization from the cell surface is clathrin-mediated endocytosis (CME) (Salisbury et al, 1980; Brown & Song, 2001; Stoddart et al, 2002). Clathrin is recruited to the BCR through its adaptor AP-2, which binds to the intracellular tyrosine-based activation motifs (ITAMs) in the CD79A and CD79B subunits (Stoddart et al, 2002; Busman-Sahay et al, 2013). In addition, antigen binding to the BCR regulates CME by Lyn-mediated phosphorylation of the clathrin heavy chain (Stoddart et al, 2002; Stoddart et al, 2005), promoting its binding to clathrin light chain and coupling clathrin-coated pits (CCPs) to the actin cytoskeleton (Bonazzi et al, 2011). Activated BCRs also associate with E3-ubiquitin ligases c-Cbl and Cbl-b, resulting in ubiquitination of the BCR CD79A and CD79B subunits, which likely plays a role in both BCR-antigen internalization and downstream intracellular trafficking (Kitaura et al, 2007). Recent proteomic studies have further

1 Immune Receptor Activation Laboratory, The Francis Crick Institute, London, UK
2 Wellcome-Wolfson Institute for Experimental Medicine, Queen's University Belfast, Belfast, UK
3 Institute of Cellular & Molecular Immunology, University Medical Center Göttingen, Göttingen, Germany
4 Institute of Immunity and Transplantation, University College London, London, UK
*Corresponding author. Tel: +44 028 9097 6461; E-mail: d.malinova@qub.ac.uk
**Corresponding author (lead contact). Tel: +44 020 3796 1609; E-mail: pavel.tolar@crick.ac.uk

substantiated extensive links between BCR signaling and CCP components (Satpathy *et al*, 2015).

However, CME is unlikely to be exclusively responsible for antigen-induced BCR internalization. For example, genetic depletion of clathrin only partly reduced BCR endocytosis, revealing another, cholesterol and actin-dependent pathway (Stoddart *et al*, 2005). Furthermore, within seconds of antigen binding, the ITAMs of the BCR are tyrosine-phosphorylated, which abrogates AP-2 binding, predicting that BCR signaling and internalization are mutually exclusive for individual receptors (Hou *et al*, 2006; Busman-Sahay *et al*, 2013). This is paradoxical, as antigen binding dramatically increases, rather than reduces, the speed of BCR endocytosis. Thus, the BCR may use several alternative mechanisms of endocytosis.

Recently, our understanding of endocytic pathways has been broadened by characterization of a clathrin-independent endocytic route termed fast endophilin-mediated endocytosis (FEME). FEME is mediated by the endophilin A family of membrane trafficking proteins, which sense and induce membrane curvature through their BAR domains (Bai *et al*, 2010; Mim *et al*, 2012). Endophilins were originally described as components of intracellular clathrin-coated vesicles in neurons (Milosevic *et al*, 2011). However, recent data from non-neuronal cells identify FEME as a clathrin-independent ligand-triggered endocytic mechanism for a range of cell surface receptors, including growth factor receptors and G-protein-coupled receptors (Boucrot *et al*, 2015; Renard *et al*, 2015; Bertot *et al*, 2018). FEME shares many requirements with CME, such as the need for actin cytoskeleton and dynamin2, making it difficult to distinguish from CME using traditional pharmacological treatments (Boucrot *et al*, 2015; Watanabe & Boucrot, 2017). However, FEME requires ligand-induced assembly of receptor signaling complexes in order to recruit endophilin A proteins to the plasma membrane via their SH3 domains (Boucrot *et al*, 2015).

In addition to FEME, endophilins have been implicated in a number of membrane remodeling processes including formation of podosomes (Sánchez-Barrena *et al*, 2012), autophagy (Soukup *et al*, 2016), and cellular polarity (Genet *et al*, 2019). Through regulation of vesicle trafficking, endophilins are also essential for maintenance of neuronal health (Milosevic *et al*, 2011; Murdoch *et al*, 2016; Soukup *et al*, 2016) and have been strongly linked to neurodegeneration in Parkinson's disease (Soukup *et al*, 2016; Nguyen *et al*, 2019). However, the molecular mechanisms of endophilin function in neurons and other cell types are still under debate.

Here, using whole-genome genetic screening in B cells we characterize the molecular components of antigen endocytosis. In addition to confirming the role of CME, we uncover a novel role for endophilin A2, the only endophilin A family member expressed in hematopoietic cells, in clathrin-independent internalization of the antigen-stimulated BCR. These findings support the growing evidence for the importance of clathrin-independent endocytosis in the function of a variety of cell surface receptors. In addition to regulating BCR endocytosis, we show that endophilin A2 is essential in activated B cells to maintain mitochondrial respiration via regulation of transferrin receptor endocytosis and iron uptake. Consequently, mice deficient in endophilin A2 show impaired expansion of GC B cells and poor production of high-affinity IgG antibodies. These results point to a selective importance of endophilin A2-mediated endocytic trafficking in GC B-cell responses.

# Results

## Improved CRISPR-based screening allows annotation of the antigen uptake pathway

To comprehensively identify genes regulating BCR-mediated antigen uptake in B cells using CRISPR screening, we generated a Ramos cell line stably expressing the *Streptococcus pyogenes* Cas9 nuclease. The human Burkitt's lymphoma line has intact tonic and ligand-induced BCR signaling and efficient endocytosis of surrogate antigen, anti-IgM F(ab')$_2$ fragments. We used second generation lentiviral vectors to transduce two pooled genome-wide CRISPR/Cas9 libraries—GeCKO[25] and Brunello (Doench *et al*, 2016). After drug selection for sgRNA expression, we screened for genetic disruptions affecting surrogate antigen endocytosis using a soluble fluorescent surrogate antigen internalization assay and flow cell sorting (Fig 1A). We extracted genomic DNA from sorted and total populations, amplified the integrated viral genomes, and analyzed the abundance of sgRNAs by next-generation sequencing. Depletion or enrichment of sgRNAs in sorted populations that did or did not internalize the surrogate antigen was expressed as a log$_2$ ratio of sgRNA abundance, called here the internalization score. This allowed us to infer the contribution of targeted genes to antigen internalization.

To assess library targeting efficiency, we have also determined a CRISPR survival score as the log2 ratio of sgRNA abundance pre- and post-drug selection. We found that survival scores of previously described essential genes (Wang *et al*, 2015) were significantly lower than survival scores of non-targeting controls (Fig EV1A and C). However, the reduction in essential gene survival scores was greater with Brunello library. Thus, large-scale gene disruption is functional with both libraries, but is more efficient with the Brunello library in line with its optimized on-target efficiency (Doench *et al*, 2016).

Analysis of gene internalization scores from GeCKO library showed that among the top hits with a positive role in antigen internalization included components of CCPs (EPN1, PICALM), intracellular trafficking (RAB7A, VPS16), regulators of the actin cytoskeleton (ACTR2, CCZ1B), exocyst components (EXOC7), and BCR signaling components (GRB2, LYN, CBL), thus establishing that the screen is detecting realistic effects (Fig 1B). Surprisingly, the screen also identified Endophilin A2 (encoded by *SH3GL1*), a critical component of FEME. We validated the key endocytic genes, including *SH3GL1*, using individual sgRNAs, as elaborated below. However, the statistical significance of the hits from the GeCKO library screen was generally low (Dataset EV1) and the validation rate of other candidates was poor (not shown). This was likely due to the low efficiency of gene targeting, supported by an inverse correlation between the predicted sgRNA on-target efficiency and survival scores calculated for the known essential genes (Fig EV1B). This was not observed for the Brunello library (Fig EV1D).

The Brunello library showed higher statistical significance for top screen hits (Fig 1C, Dataset EV2). The hits included regulators of endocytosis (EPN1, EPS15, PICALM), as well as genes predicted to be involved in a wide range of cellular functions. To validate gene candidates in a high-throughput manner, we re-screened using a custom CRISPR sgRNA minilibrary containing three sgRNAs per gene for the top 213 positive or negative regulators, which were also highly expressed in Ramos cells (Dataset EV3). In addition, the

custom minilibrary contained 15 essential genes as positive controls, and 48 non-targeting sgRNA controls. Analysis of essential gene targeting showed efficient genetic disruption by the custom library (Fig EV1E). The internalization re-screen validated the function of 72 genes with strong statistical significance ($q < 0.05$; Fig 1D, Dataset EV3), including 58 positive regulators (score $< -0.4$) and 14 negative regulators (score $> 0.35$), covering a broad range of

cellular processes such as endocytosis, endosomal and lysosomal trafficking, ubiquitination, actin cytoskeleton, Golgi and ER biology, and transcriptional regulation (Fig 1E).

Together, these experiments represent the first comprehensive analysis of the genetic make-up of the BCR internalization pathway and strongly implicate specific endocytic genes as well as downstream feedback from a large and complex regulatory network.

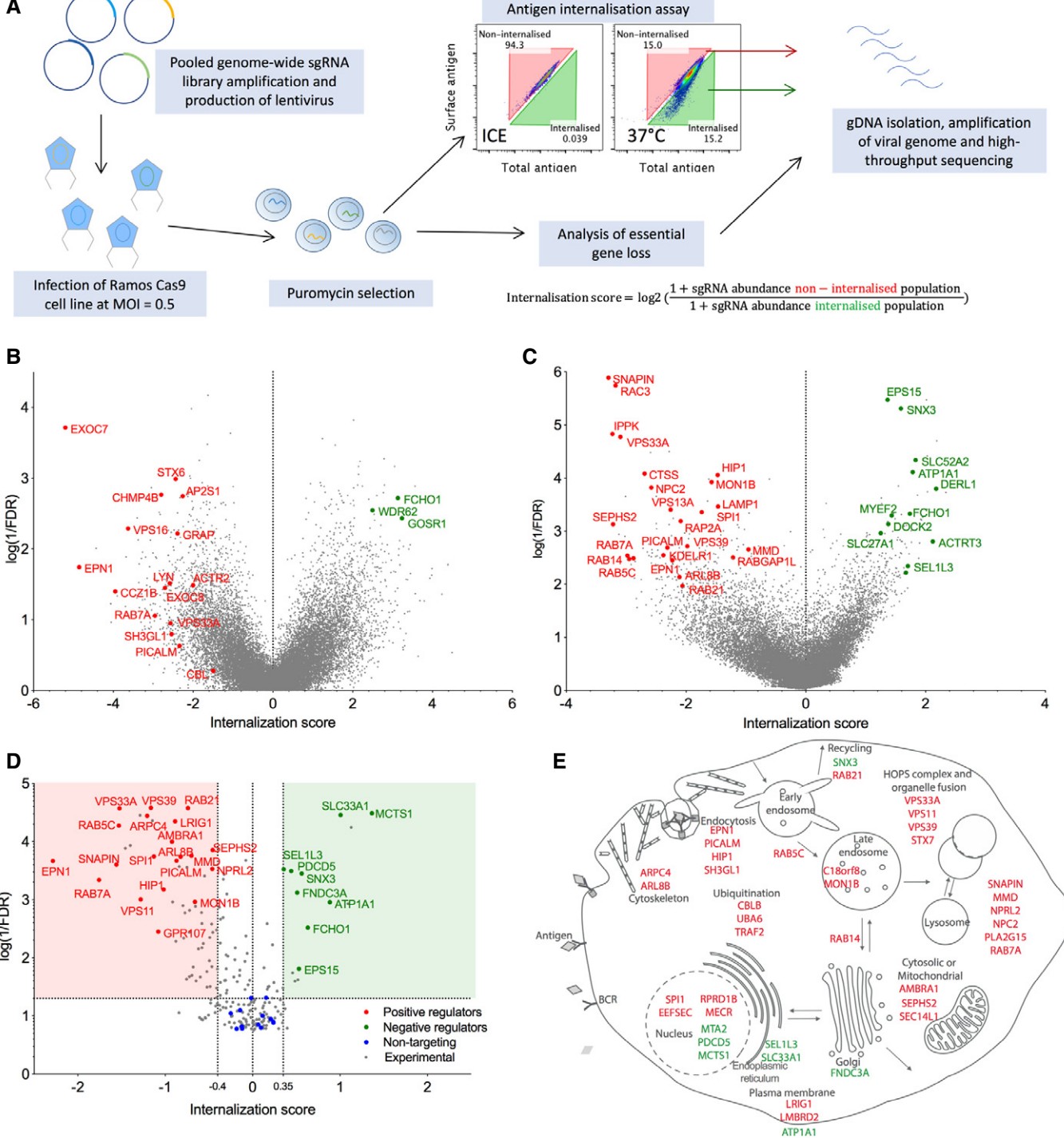

**Figure 1.**

◀

**Figure 1.  CRISPR/Cas9 screens identify regulators of B-cell antigen internalization.**

A   Genome-wide screen and internalization assay workflow. Library-transduced Ramos Cas9 cells were incubated with fluorescent, biotinylated surrogate antigen, anti-IgM F(ab')$_2$ (total antigen) and subsequently stained with fluorescent streptavidin (surface antigen). Flow cytometry plots show a control sample incubated on ice preventing anti-IgM F(ab')$_2$ endocytosis and an experimental sample from which cells displaying high and low anti-IgM F(ab')$_2$ internalization were flow sorted and analyzed for gRNA abundance.
B   GeCKO library results showing mean internalization score per gene versus statistical significance. Whole-genome data come from two independent repeats.
C   Brunello library results. Whole-genome data come from two independent repeats.
D   Minilibrary validation results from four independent experiments. Validated candidates with internalization score < −0.4 or > 0.35 and q < 0.05.
E   Selected validated candidates grouped by functional annotation.

Data information: In (B–D), genes with annotation related to endocytosis or intracellular trafficking are highlighted. Red denotes examples of positive regulators; green denotes examples of negative regulators of internalization. Full results can be found in Datasets EV1–EV3.

## Endophilin A2 is involved in antigen-dependent BCR internalization

To understand the role of candidate genes on antigen uptake mechanistically, we focused on genes predicted to directly regulate endocytosis. In addition to a subset of CCP components, such as epsin 1 (EPN1), PICALM, and HIP1, the screen implicated endophilin A2, suggesting a potential role of clathrin-independent endocytosis. We validated the involvement of the key genes by individual CRISPR-mediated knockouts in Ramos Cas9 cells (Fig 2A and B) and in primary B cells (Fig 2C). As endophilin A2 has not been studied in B cells before, we focused on its role in subsequent studies. In both Ramos cells and primary mouse B cells, targeting of *SH3GL1* reduced soluble surrogate antigen uptake to an extent comparable with targeting CME (Fig 2B and C). *SH3GL1* loss also strongly impaired internalization of surrogate antigen from immune synapses with membrane substrates (Fig 2D). However, endophilin A2 was important only for internalization of an activating anti-IgM F(ab')2 fragment, but not for a non-stimulatory anti-IgM Fab fragment, whereas CME regulated internalization of both (Fig 2E). Thus, endophilin A2 in an important novel player selectively involved in BCR internalization upon antigen ligation.

To determine whether endophilin A2 regulates BCR endocytosis directly, we overexpressed endophilin A2-GFP in Ramos cells and imaged its localization in cells forming immune synapses with anti-IgM-coated plasma membrane sheets. Endophilin A2 formed highly dynamic spots in the synapse that colocalized with BCR clusters (Fig 2F, Movie EV1). Furthermore, imaging of endophilin A2-GFP and mCherry-clathrin light chain suggested that endophilin A2 works independently of CCPs as the majority of endophilin A2-GFP spots did not colocalize with CCPs labeled with mCherry-clathrin (Fig 2F). Quantification confirmed that only 17.44% of endophilin A2 clusters overlapped with CCPs and only 15.65% of CCPs overlapped with endophilin A2 spots, only slightly higher than colocalization in randomized controls (Fig 2G). In addition, live cell imaging showed endophilin A2-GFP spots interacted with BCR clusters without prior or subsequent recruitment of clathrin (Movie EV1). To confirm clathrin-independent colocalization of endophilin A2 with the BCR, we quantified colocalization of surrogate antigen clusters with endophilin A-GFP and mCherry-clathrin. This analysis showed that around 15% of BCR clusters localized to FEME spots, and a similar percentage localized to CME; by contrast, only 3% of BCR clusters were associated simultaneously with both markers, strongly suggesting independent roles of the two endocytic pathways (Fig 2H). Finally, disruption of CME-mediated BCR endocytosis following deletion of clathrin adaptors EPN1 or PICALM1 did not

significantly affect colocalization of surrogate antigen clusters with endophilin A2 (Fig 2I). These observations are in line with a clathrin-independent role of endophilin A2 in FEME.

To understand how endophilin A2 is recruited to activated BCRs, we analyzed endophilin A2 complexes in anti-IgM-stimulated Ramos cells using a BioID2 proximity labeling assay (Kim *et al*, 2016) (Fig EV2A). This assay captured known interaction partners of Endophilin A2, including dynamin2 (*DNM2*), intersectin2 (*ITSN2*) (Burbage *et al*, 2018a), and Alix (*PDCD6IP*) (Mercier *et al*, 2016). Among the detected interaction partners were also BCR signaling adaptors GRB2 and BLNK (Fig EV2B, Dataset EV4). To test the importance of these adaptors, we used pull-down assays to analyze interactions of endophilin A2-GFP with the BCR in DG75 Burkitt lymphoma cells in the context of *GRB2* or *BLNK* deletion (Vanshylla *et al*, 2018). Similar to Ramos cells, endophilin A2-GFP was eluted in the anti-IgM-interacting fraction in WT DG75 cells (Fig 2J). The amount of endophilin A2-GFP was significantly reduced in anti-IgM pulldown in the absence of GRB2 or BLNK. Furthermore, while we detected similar numbers of endophilin A2-GFP spots in synapses of GRB2 and BLNK-deficient DG75 cells compared with parental DG75 cells, deletion of *GRB2* significantly reduced endophilin A2 colocalization with surrogate antigen clusters (Fig 2K). These results indicate that endophilin A2's recruitment to ligand-activated BCRs is regulated by complexes organized by the adaptor GRB2 with possible participation of BLNK.

## Endophilin A2 deletion results in abrogated development and function of mature B cells

Endophilin A2 is the only endophilin A family member expressed in hematopoietic cells. To understand the role of Endophilin A2 in immune cell development and responses to antigens *in vivo*, we developed a CRISPR gene targeting strategy in mouse hematopoietic stem cells (HSCs) followed by adoptive transfer to irradiated host mice (Fig 3A). We used HSCs from Cas9 knockin mice (expressing GFP) and targeted them with lentivirus encoding sgRNA and a mCherry marker. In the resulting chimeric mice, CRISPR-targeted cells were detected by flow cytometry and mCherry[+] percentage of the donor GFP[+] population was used to monitor cell development in the bone marrow and in the spleen. As a control, we used a *Cd4*-targeting sgRNA, which showed efficient abrogation of the development of mCherry[+] CD4-positive T cells, but did not affect mCherry[+] proportions throughout B-cell development (Fig 3B). Thus, this strategy is efficient and specific for gene targeting *in vivo* and allows analysis of gene effects on B-cell development and immune responses.

Targeting with a sgRNA specific for *Sh3gl1* showed an efficient disruption of *Sh3gl1* at the genomic level (Fig EV3A) and a reduction in *Sh3gl1* mRNA (Fig EV3B). *Sh3gl1*-targeted mCherry$^+$ B-cell proportions in the bone marrow and in the transitional populations in spleen remained constant and similar to controls, indicating that early B-cell development progressed normally in the endophilin A2-deleted cells (Fig 3B). However, proportions of mCherry$^+$ cells declined to about half in mature follicular B cells and were 5–10 times reduced in marginal zone (MZ) B cells (Fig 3B and C). In comparison, mCherry proportions in T cells (Fig 3B) and other non-B cells (not shown) were not affected, indicating a selective role of endophilin A2 in B cells.

To investigate antigen-specific responses of *Sh3gl1*-targeted B cells to immunization, we crossed Cas9 mice with SW$_{HEL}$ mice (Phan *et al*, 2003) carrying B-cell specificity for hen egg lysozyme (HEL). Bone marrow from the resulting progeny was used for CRISPR targeting and bone marrow chimeras. Following adoptive transfer of follicular B cells to C57BL/6 mice, we immunized with

HEL conjugated to sheep red blood cells (SRBCs) and analyzed early activation, GC responses, and generation of plasma cells (PCs). At day 1 after immunization, *Sh3gl1*-targeted B cells upregulated activation markers CD69 and CD86 similarly to controls (Fig 3D and E). However, at days 4–14, they produced significantly fewer class-switched IgG1 cells, GC cells, and PCs (Fig 3F). Naïve follicular B-cell numbers also showed a decreasing trend but did not reach significance. Thus, endophilin A2 is dispensable for early antigen-induced B-cell activation, but is required for GC responses and efficient generation of PCs.

### Endophilin A2 knockout mouse confirms B-cell developmental defects and exhibits diminished GC and antibody responses

To confirm the importance of endophilin A2 in B-cell responses, we characterized the immune system of *Sh3gl1*$^{-/-}$ mice (Milosevic *et al*, 2011) compared with *Sh3gl1*$^{+/+}$ littermates (WT). In agreement with the CRISPR-mediated deletion, *Sh3gl1*$^{-/-}$ mice showed

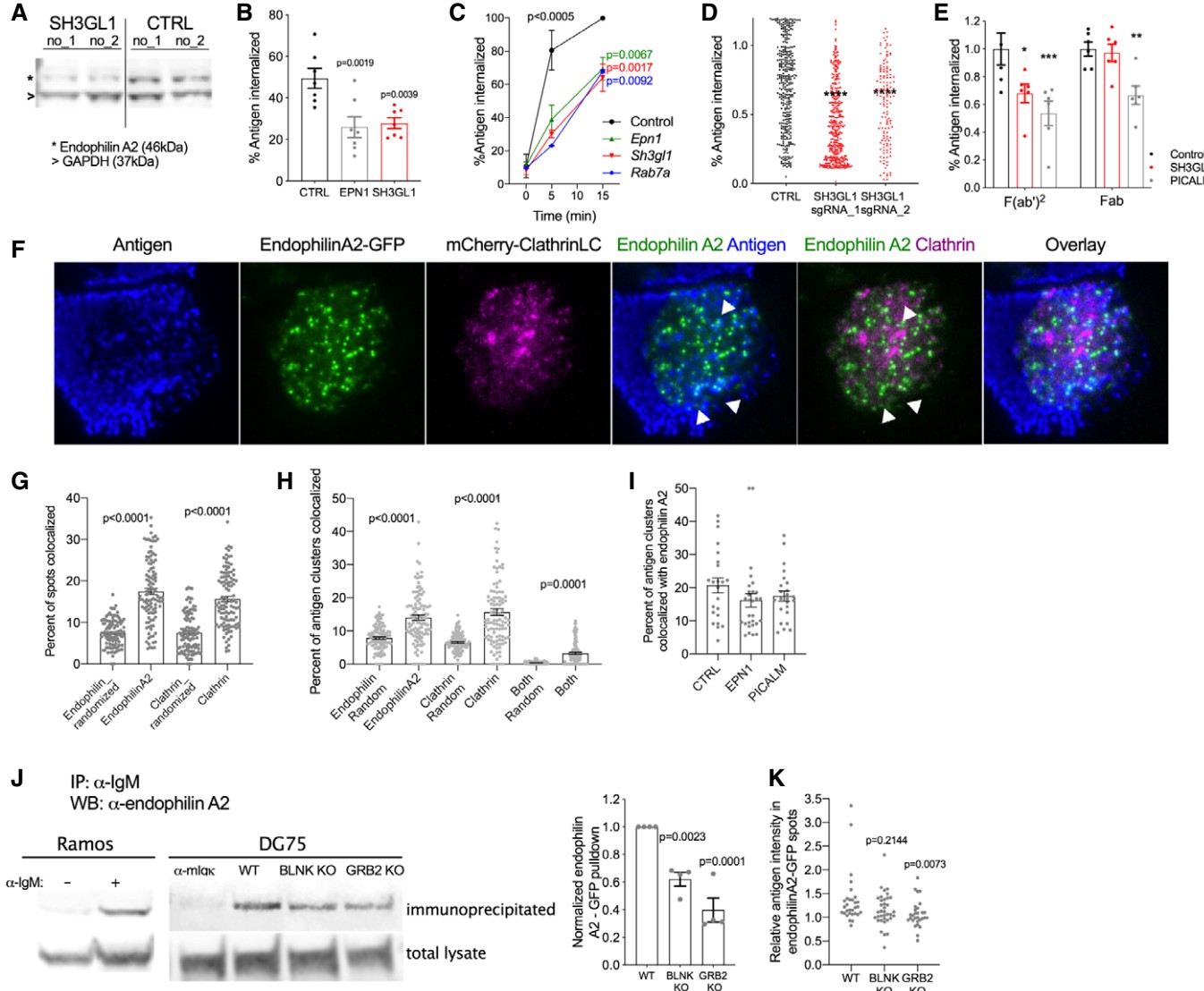

**Figure 2.**

◀

**Figure 2. Endophilin A2 regulates antigen-induced BCR internalization in a clathrin-independent manner.**

A   Western blot showing loss of endophilin A2 protein after *SH3GL1* targeting in Ramos cells using 2 independent sgRNAs, compared with 2 non-targeting sgRNAs.
B   Soluble anti-IgM F(ab')$_2$ antigen internalization at 20 min in Ramos cells targeted with indicated sgRNAs. *P*, significance in one-way ANOVA versus control. Plot shows mean and SEM of 7 independent sgRNA infections.
C   Validation of gene candidates affecting soluble antigen uptake in primary mouse B cells. Follicular B cells isolated from CRISPR-targeted bone marrow chimera mice internalize anti-mouse IgM F(ab')$_2$. Data show mean and SEM of *N* = 3 mice. *P*, statistics from 2-way ANOVA; at 5 min, all 3 genotypes show *P* < 0.0005 versus control; at 15 min *P* values are indicated in corresponding color.
D   Antigen internalization from PMS in *SH3GL1*-targeted Ramos cells. ****P < 0.0001 in one-way ANOVA. *N* = 547, 389 and 136 cells, respectively, from 2 experiments.
E   Internalization of soluble Fab or F(ab')$_2$ fragments in Ramos cells targeted with indicated sgRNAs. *P = 0.0124 **P = 0.0092 ***P = 0.0004 using 2-way ANOVA. Data show mean ± SEM, from 6 biological replicates across two experiments.
F   TIRF images of Ramos cells transduced with endophilin A2-GFP and mCherry-ClathrinLC on antigen-loaded PMS. Scale bar = 5 μm. White arrowheads highlight endophilin A2 colocalization with antigen clusters. See also Movie EV1.
G   Percent of endophilin A2 and clathrinLC spots colocalizing with each other. Levels of random spot colocalization were calculated using randomized spot coordinates. Data show mean and SEM from 107 cells analyzed from 4 experiments; statistics from one-way ANOVA.
H   Percentage of surrogate antigen spots colocalizing with endophilin A2, clathrin LC or both markers. Levels of random spot colocalization were calculated using randomized spot coordinates. Data show mean and SEM from 107 cells analyzed from 4 experiments; statistics from one-way ANOVA.
I   Colocalization of surrogate antigen clusters with endophilin A2-GFP following deletion of clathrin adaptors EPN1 or PICALM1. Data show mean and SEM from 23–29 cells.
J   IgM-BCR immunoprecipitation to investigate recruitment of endophilin A2 to activated BCRs. Anti-endophilin A2 blots detect overexpressed endophilin A2-GFP in streptavidin immunoprecipitates and total supernatants from anti-IgM-biotin-stimulated Ramos and DG75 cells. Representative experiment and quantification from three independent pulldowns in WT, *BLNK*-, or *GRB2*-knockout (KO) DG75 cells. Pulldown intensity is calculated as a percentage of total endophilin A2 protein and normalized to WT in each experiment. Data show mean and SEM; *P* values calculated using one-way ANOVA.
K   Recruitment of endophilin A2 to antigen clusters quantified as mean antigen intensity in endophilin A2 clusters relative to mean antigen intensity in the entire synapse. *N* = 30, 29, and 36 cells, respectively, from 2 experiments. *P*, significance in Kruskal–Wallis test with multiple comparisons.

normal early B-cell development in the bone marrow and in the transitional populations in the spleen but a reduction in splenic follicular and MZ B cells, as well as mature B cells in lymph nodes (Fig 4A). In contrast, numbers of CD4$^+$ and CD8$^+$ T cells in lymph nodes were normal. This B-cell lymphopenia was reflected in reduced spleen size of *Sh3gl1*$^{-/-}$ mice, compared with WT littermates (Fig 4B). In addition, *Sh3gl1*$^{-/-}$ follicular B cells had an altered surface marker phenotype, similar to that obtained by CRISPR targeting (Fig 4C, and not shown).

To understand the role of endophilin A2 in antibody production, we immunized the mice with T-dependent antigen, NP-CGG. *Sh3gl1*$^{-/-}$ mice exhibited severely reduced total and NP-specific GC B-cell numbers (Fig 4D). Quantification of total and NP-specific serum antibody levels showed that *Sh3gl1*$^{-/-}$ mice had normal total baseline IgM and IgG levels (Fig EV3C–F) and mounted a relatively normal IgM response (Fig 4E). However, they showed markedly decreased serum NP-specific IgG1 and IgG3 at 7 and 14 days post-immunization (Fig 4F).

To investigate the effect of endophilin A2 knockout on affinity maturation, we compared the binding of serum antibodies with low valency NP7 and high valency NP25 antigens. The NP7/NP25 binding ratio, used as a measure of affinity, increased in WT mice by day 7 post-immunization. While no differences were seen in the NP7/NP25 binding ratio of IgM in *Sh3gl1*$^{-/-}$ mice, the serum IgG ratio was significantly lower at day 7 and did not reach WT levels at day 14, suggesting inefficient antibody affinity maturation (Fig 4G).

These data, together with normal numbers of T follicular helper (Tfh) cells after immunization (Fig 4H), indicate a B cell-intrinsic role of endophilin A2 in GC B-cell responses, class switching, and affinity maturation.

**Endophilin A2 is dispensable for antigen presentation to T cells**

The GC defect could reflect an inability of the endophilin A2-deleted B cells to compete for T cell help as a result of lower BCR-mediated antigen uptake, processing, and presentation. Follicular B cells isolated from the chimeric mice exhibited a small but consistent reduction in IgM-BCR internalization (Fig 5A). A similar reduction was observed in *Sh3gl1*$^{-/-}$ mice (not shown). In contrast, internalization of IgD BCR appeared unaffected in the absence of endophilin A2 (Fig 5B). In addition, we observed a significant reduction in IgG-BCR internalization after anti-IgG stimulation of *Sh3gl1*$^{-/-}$ GC B cells (Fig 5C), confirming a role for endophilin A2 in IgM and IgG-BCR endocytosis in both naïve and GC B cells. Unexpectedly, despite slower internalization after anti-IgM stimulation, both in solution (Fig 5A) and from PMS (Fig 5D), surface levels of IgM were lower in *Sh3gl1*$^{-/-}$ B cells (Fig 5E). This could be explained by increased rates of degradation of surface IgM 3 h after anti-IgM stimulation (Fig 5F and G), suggesting that while endophilin A2 positively regulates BCR endocytosis, it negatively regulates BCR delivery into degradative compartments.

To directly test the role of endophilin A2 in antigen presentation, we adoptively transferred HEL-specific B cells from endophilin A2 HSC-targeted bone marrow chimeras into WT mice, followed by immunization with SRBC-HEL conjugated to Ea peptide, as previously described (Brink *et al*, 2015). B-cell antigen presentation was analyzed 1 day later by staining for Eα peptide-MHCII complex on the cell surface. *Sh3gl1*-targeted B cells exhibited a very small reduction in antigen presentation, despite lower antigen uptake (Fig 5H and I). Similar results were obtained *in vitro* when isolated B cells were cultured with Eα peptide conjugated to anti-Igκ via a biotin-streptavidin bridge (Fig 5J). Furthermore, sorted mCherry$^+$ *Sh3gl1*-targeted B cells pulsed with anti-Igκ conjugated to ovalbumin stimulated normal proliferation of ovalbumin-specific OT-II CD4 T cells in co-culture assays (Fig 5K). Total antigen presentation was also similar in GC from wild-type or *Sh3gl1*$^{-/-}$ immunized mice (Fig 5L). Thus, despite the role in BCR internalization, endophilin A2 is not essential for total B-cell antigen presentation to T cells. This may be explained by a compensatory mechanism, as endophilin A2 simultaneously limits degradative BCR trafficking and MHCII expression.

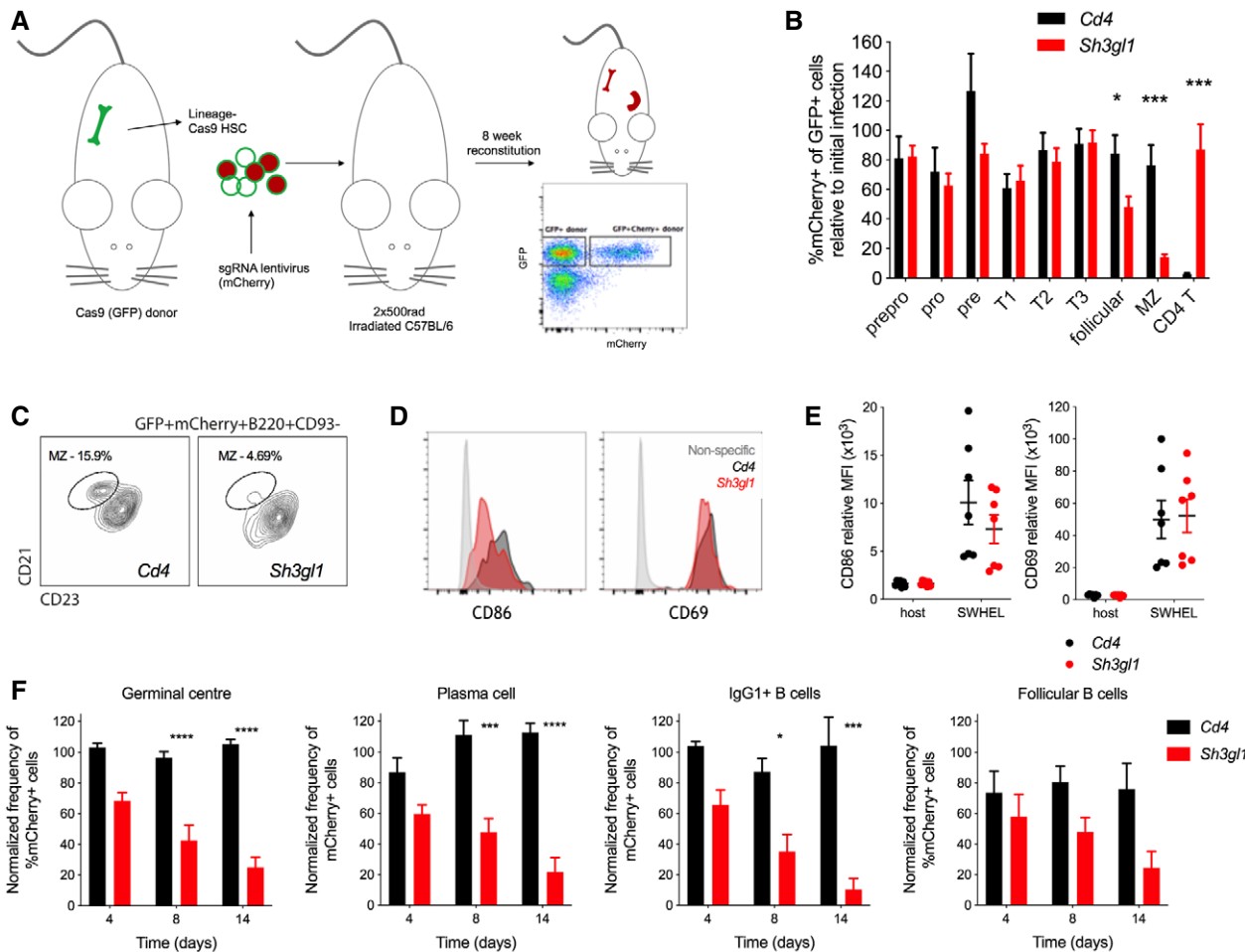

**Figure 3. Endophilin A2 is required for peripheral B-cell development and GC responses.**

A  CRISPR-targeted bone marrow chimera workflow, showing flow cytometry detection of donor-derived (GFP⁺), sgRNA-containing (mCherry⁺) cells.
B  Development of B-cell populations in chimeras' bone marrow and spleen from *Cd4*- or *Sh3gl1*-targeted HSCs. Percentage of mCherry⁺ cells at each stage is normalized to initial infection rate of transferred lineage⁻ cells, and shown as mean + SEM. *N* = 11–12 mice. *P = 0.0286, ***P < 0.0001 using two-way ANOVA.
C  MZ development in mCherry⁺ B-cell compartment.
D  Representative stain of surface CD86 and CD69 on adoptively transferred HEL-specific follicular B cells 24 h post-SRBC-HEL immunization.
E  Quantification of surface CD86 and CD69 mean fluorescence intensity (MFI). Data show mean and SEM from six mice across two experiments.
F  Fractions of mCherry⁺ cells in GC, PC, IgG1⁺ class-switched and follicular B cells at 4-, 8-, and 14 days post-SRBC-HEL immunization. *N* = 8–10 mice; data shows mean and SEM. *P < 0.05; ***P < 0.001; ****P < 0.0001 using 2-way ANOVA with multiple comparisons.

We also observed normal antigen-induced calcium flux (Fig EV3G and H), comparable levels of Syk, Erk, and Akt phosphorylation downstream of the BCR (Fig 3I) and chemokine-induced migration (Fig EV3J), suggesting that early functions required for B-cell activation and interaction with T cells *in vivo* are not affected by endophilin A2 absence.

## Endophilin A2 is important for metabolic support of B-cell growth and proliferation

To understand whether endophilin A2 is required for the response of B cells to T cell help, we analyzed proliferation of DDAO-labeled B cells stimulated by CD40L. We found a severely reduced ability of *Sh3gl1*-targeted B cells to expand in response to CD40L compared with controls (Fig 6A). However, CD40 expression in *Sh3gl1*⁻/⁻ B

cells was normal and early transcription induced by CD40L was also similar to WT B cells as evidenced by RNAseq (Figs 6B and EV4A, Dataset EV5), suggesting a more downstream defect in cell expansion. Consistently, we observed that the defect in expansion was not limited to CD40L stimulation, but also occurred in response to BCR and TLR stimuli (Fig 6C). The reduced expansion was primarily a result of reduced proliferation, as rates of cell death in response to CD40, BCR, and TLR ligation were not elevated (Fig 6D). However, cell survival in response to BAFF was reduced. These findings were reproducible in non-competitive B-cell cultures from *Sh3gl1*⁻/⁻ mice (Fig EV4B and C), although results from IgM and TLR stimulation did not reach statistical significance under these conditions. In contrast, T cells isolated from these mice proliferated similarly to WT in response to anti-CD3 plus anti-CD28 stimulation (Fig EV4B and C).

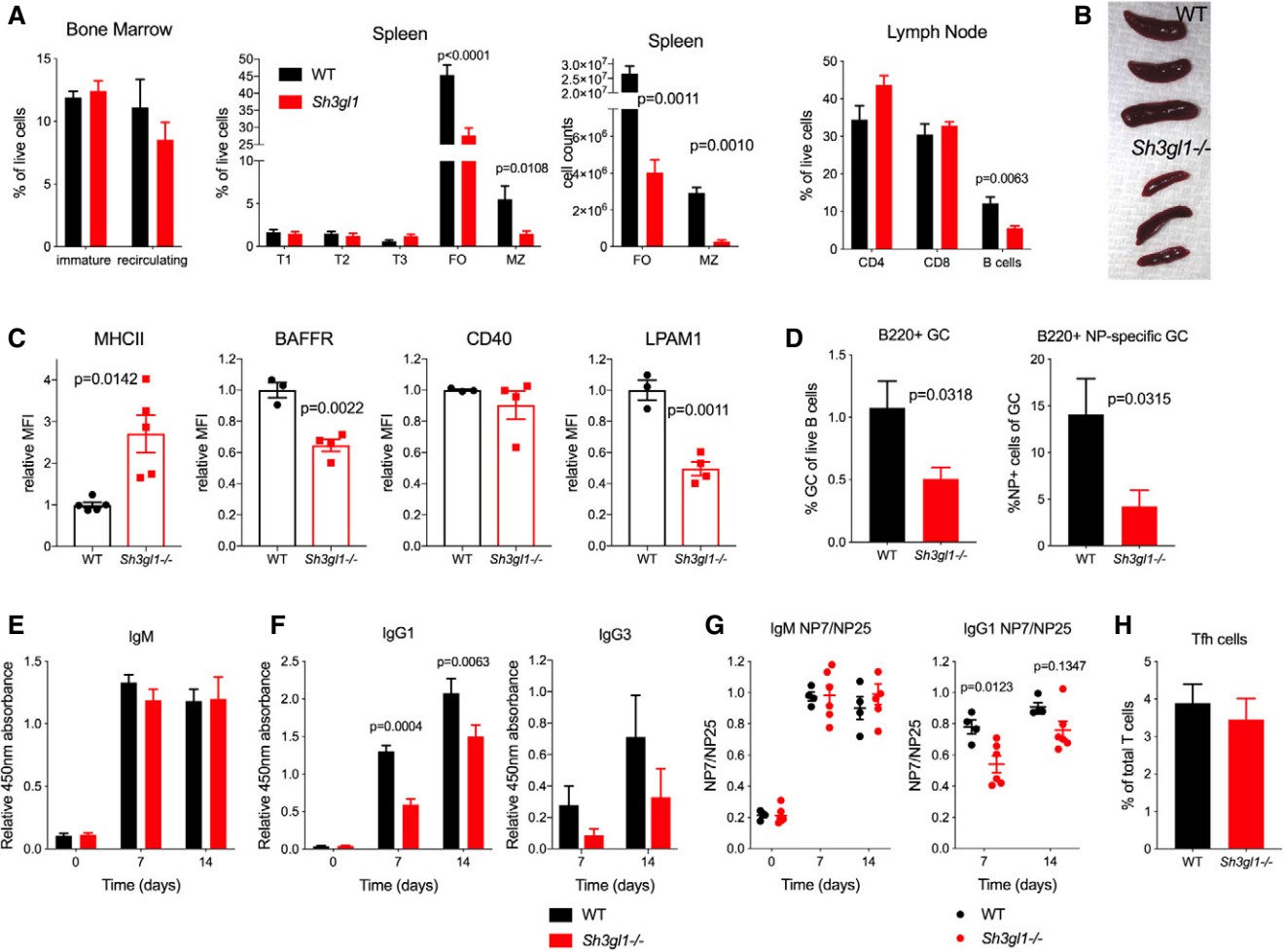

**Figure 4. Endophilin A2 knockout mouse mimics the phenotype of CRISPR-targeted chimeras and exhibits reduced levels of affinity matured serum antibodies upon immunization.**

A   Characterization of B-cell populations in bone marrow, spleen, and lymph node at steady state. Data show mean and SEM from $N = 5$ mice. $P$, significance in $t$-tests performed for follicular (FO) and MZ cell counts.

B   Spleen size in $Sh3gl1^{-/-}$ and WT littermates at 12 weeks of age.

C   Surface levels of MHCII, BAFFR, CD40, and LPAM1 in $Sh3gl1^{-/-}$ and WT littermates. $N = 4–5$ mice. $P$, statistical significance from unpaired $t$-tests. Data show mean ± SEM.

D   Quantification of B220$^+$Fas$^+$CD38$^-$ GC population as percentage of total splenic B cells and NP-specific cells as a percentage of the GC population 14 days after NP-CGG immunization in alum. Data show mean ± SEM from $N = 6$ mice, $P$, significance in unpaired $t$-tests.

E, F   Serum antibodies collected pre-immunization and 7 or 14 days post-NP-CGG immunization, detected by NP7-BSA ELISA using isotype-specific secondary antibodies. Data show mean ± SEM from $N = 3$ experiments with statistical significance from 2-way ANOVA with multiple comparisons.

G   Ratio of binding to NP7 and NP25, as measured by ELISA. Data show mean ± SEM; $N = 4–6$ mice. Statistical significance from 2-way ANOVA with multiple comparisons.

H   Numbers of Tfh cells 14 days post-immunization, identified as a CD4$^+$CD44$^+$PD-1$^+$CXCR5$^+$ population. Mean and SEM of $N = 4$ mice.

Analysis of CRISPR survival scores determined in Ramos cells using the Brunello library also indicated cell-intrinsic essentiality of endophilin A2 in growth (Fig EV4D). In addition, re-analysis of published CRISPR screens (Wang *et al*, 2015; Wang *et al*, 2017; Phelan *et al*, 2018), indicated selective requirement for endophilin A2 for growth of B-cell lymphoma lines, particularly Burkitt and GCB diffuse large B-cell lymphoma types, but not the diffuse large B-cell ABC type or other hematopoietic lines (Fig EV4D). In contrast, critical components of CME, EPN1 and PICALM, were not

essential in Ramos cells or most other lymphoma cell lines (Fig EV4D).

CRISPR-mediated *SH3GL1* targeting in Ramos cells confirmed a progressive loss of the targeted (mCherry$^+$) population over time (Fig 6E), which was further exacerbated with increased levels of competition (Fig EV4E). This cell-intrinsic defect was also apparent in puromycin-selected cultures as reduced cell numbers at each passage and was specific to endophilin A2 as compared to CCP components (Fig 6F). Using a BrdU pulse-chase assay in *SH3GL1*-

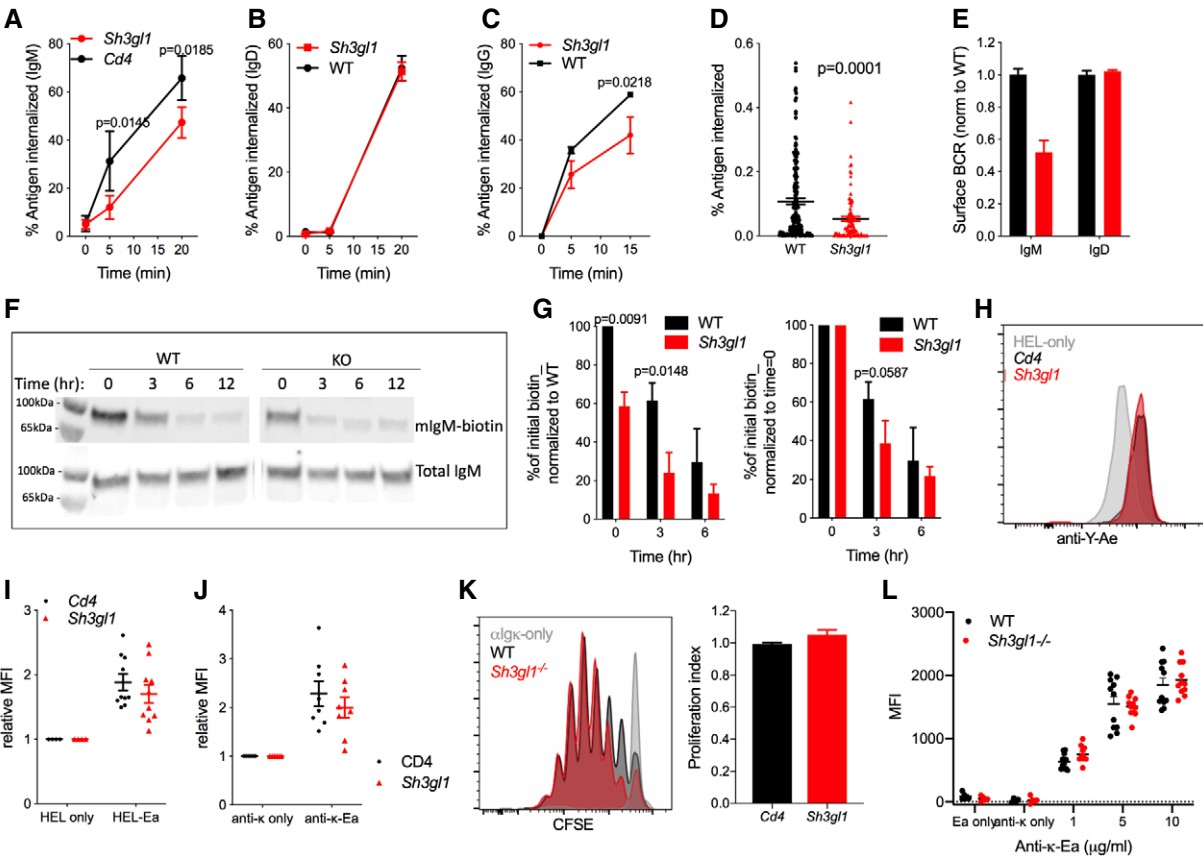

**Figure 5. Endophilin A2 positively regulates BCR internalization but is dispensable for antigen presentation.**

A   Soluble anti-IgM internalization in CRISPR-targeted follicular B cells. N = 11–12 mice. Data show mean ± SEM. P, statistical significance from two-way ANOVA.

B   Soluble anti-IgD internalization. N = 3 mice. Data show mean ± SEM. P, statistical significance from two-way ANOVA.

C   Soluble anti-IgG internalization in GC B cells from SRBC-immunized mice. N = 4 mice. Data show mean ± SEM. P, statistical significance from two-way ANOVA.

D   Internalization of anti-IgM from PMS quantified in fixed naïve B cells at 10-min post-synapse formation. N = 172 and 97 cells analyzed in one out of two representative experiments. Data show mean ± SEM. P = 0.0001 in unpaired t-test.

E   Surface IgM or IgD MFI relative to WT in naïve splenic follicular B cells. N = 3 mice. Data show mean ± SEM.

F   IgM degradation following total cell surface biotinylation, measured as loss of biotin-labeled membrane IgM (mIgM) in cell lysates over time.

G   Western densitometry quantifying IgM degradation assay. Proportion of biotin detected is normalized to the initial WT condition (left) or to the initial amount for each genotype (right). Data show mean ± SEM from three independent tests. P, statistical significance from two-way ANOVA.

H   MHCII antigen presentation in SWHEL B cells 24 h post-adoptive transfer and immunization with Ea-HEL-SRBC conjugate as detected by anti-Y-Ae surface stain.

I   Relative anti-Y-Ae MFI in CRISPR-targeted B cells normalized to HEL-only immunized mice. N = 10 immunized mice across 3 experiments. Data show mean ± SEM.

J   In vitro antigen presentation in CRISPR-targeted B cells measured by anti-Y-Ae. N = 8 independent cultures in 2 experiments. Data show mean ± SEM.

K   Proliferation (CFSE dilution) of OT-II CD4 cells in co-cultures with anti-Igκ-OVA pulsed B cells; and quantification from 6–8 mice. Data show mean ± SEM.

L   Relative anti-Y-Ae MFI in GC from immunized wild-type and Sh3gl1$^{-/-}$ mice. N = 11 from two experiments. Data show mean ± SEM.

targeted Ramos cells, we established that the reduced cell expansion was not a result of a block at a specific cell cycle checkpoint, but an overall slower rate of cell cycle progression (Fig EV4F).

To identify the reason for the B-cell growth defect, we performed RNA sequencing on sorted follicular B cells from WT or Sh3gl1$^{-/-}$ mice ex vivo or post-CD40L activation. Gene set enrichment analysis in the absence of endophilin (GSEA, Fig EV4G) revealed downregulation of pathways regulating cell activation, vesicle-mediated transport, endocytosis, protein transport, and signal transduction. Interestingly, gene sets upregulated in the absence of endophilin included many metabolic processes, cell cycle and organization of mitochondria, ER, and the plasma membrane. To investigate possible role of cell metabolism and

mitochondrial function, we measured oxygen consumption rate (OCR) in Sh3gl1$^{-/-}$ primary B cells and CRISPR-targeted Ramos cells. Endophilin A2-deficient cells exhibited reduced basal and maximal respiration compared with controls, consistent with metabolic impairment highlighted in the transcriptional analysis (Fig 6G and H). Contrastingly, OCR of T cells isolated from Sh3gl1$^{-/-}$ mice were similar to WT T cells (Fig 6I).

The transcriptional analysis also revealed significant enrichment (at FDR < 25%) of several iron and other metal transport and homeostasis gene sets in Sh3gl1$^{-/-}$ cells. Expression of several genes involved in regulating iron ion homeostasis was elevated in Sh3gl1$^{-/-}$ B cells (Fig EV4H). Given the importance of cellular iron homeostasis for mitochondrial respiration and lymphocyte

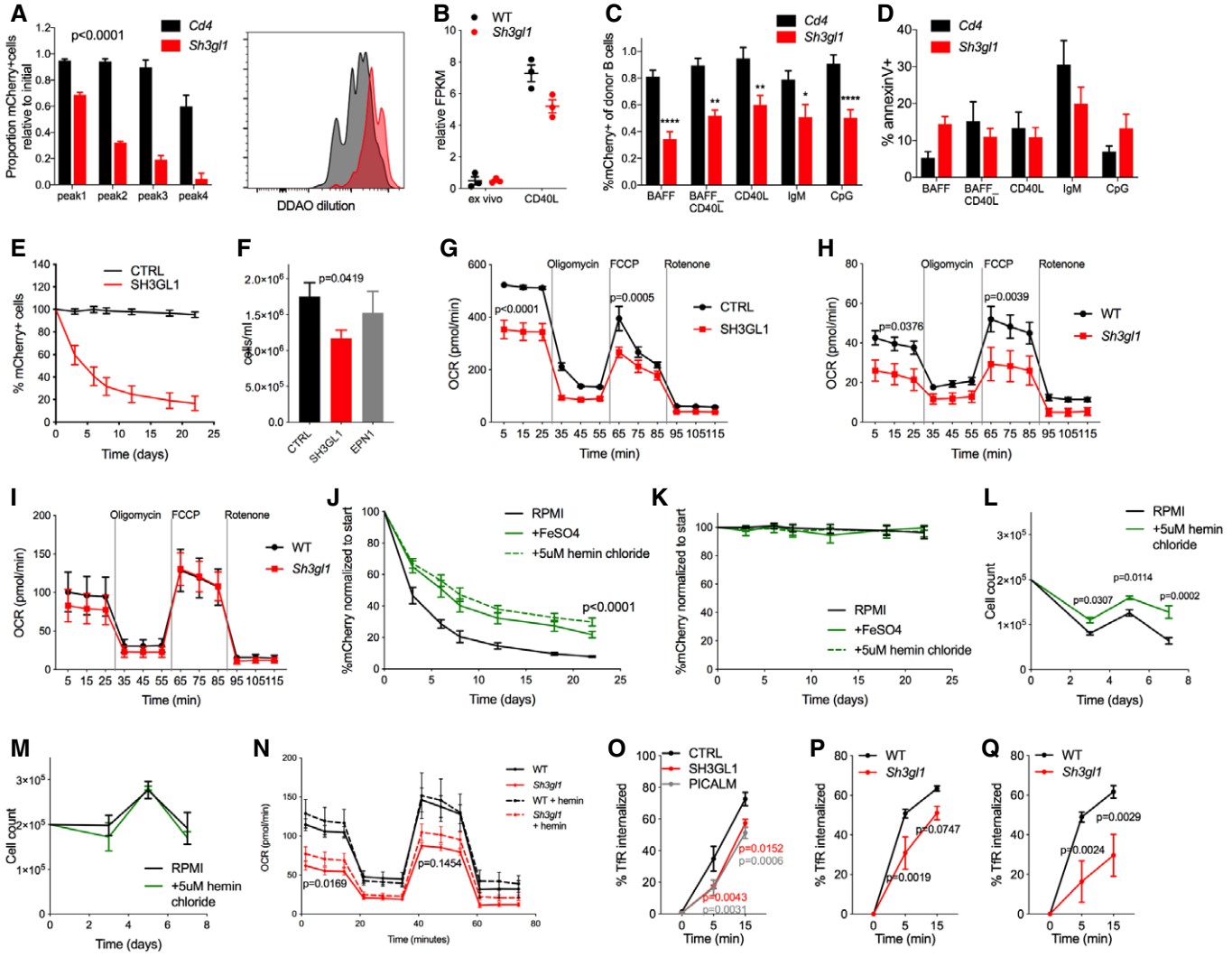

**Figure 6. Endophilin A2 is required for B-cell iron-dependent oxidative phosphorylation and cell expansion.**

A  CRISPR-targeted follicular B-cell proliferation after 3 days of CD40L stimulation. mCherry+ populations were quantified within individual DDAO peaks. N = 3 mice. P, statistical significance calculated by two-way ANOVA reached P < 0.0001 for all proliferation peaks.

B  CD80 mRNA levels in B cells following 24 h of CD40L culture, relative to untreated B cells. Data from RNAseq of N = 3 mice.

C, D  Isolated naïve B cells from CRISPR-targeted chimeras after 3-day culture in stated cytokines. (C) mCherry percentage in targeted primary cell cultures. N = 15 mice. *P < 0.05, **P < 0.01, ****P < 0.0001 using two-way ANOVA. (D) AnnexinV surface stain in targeted cells. N = 3–4 mice.

E  mCherry+ cell percentage in CRISPR-targeted Ramos cells over time. N = 6 independent experiments.

F  Cell counts at passage 3–6 of CRISPR-targeted puromycin-selected Ramos cells. N = 4 independent infections. P, statistical significance from unpaired t-test.

G  Oxygen consumption rate (OCR) in CRISPR-targeted Ramos cells. N = 2 independent infections. P, two-way ANOVA of basal and maximum OCR.

H  OCR in B cells isolated from *Sh3gl1*−/− and WT littermates. N = 9 mice. P, two-way ANOVA of basal and maximum OCR.

I  OCR in T cells isolated from *Sh3gl1*−/− and WT littermates. N = 4 mice.

J–L  Cell cultures with full RPMI media supplemented with 50 μg/ml FeSO4 or 5 μM hemin chloride. (J) *SH3GL1*-targeted Ramos cells. N = 4 independent infections. P, statistical significance from two-way ANOVA from both treatments versus RPMI. (K) Control sgRNA-targeted Ramos cells. (L) *Sh3gl1*−/− isolated B cells, maintained in CD40L culture. N = 3 mice. P, significance calculated by two-way ANOVA.

M  B cells isolated from WT littermates, maintained in CD40L culture.

N  OCR in *Sh3gl1*−/− or WT B cells after overnight culture in CD40L, supplemented with 5 μM hemin chloride. N = 6 mice. P, two-way ANOVA with matched samples for basal and maximum OCR.

O  TfR internalization assay in Ramos cells using biotinylated anti-CD71. Four independent CRISPR infections in two experiments. P, significance calculated using two-way ANOVA and indicated in corresponding color.

P, Q  Mouse transferrin internalization in B cells from SRBC-immunized WT and *Sh3gl1*−/− littermates. N = 7 mice per genotype in two experiments. P, statistical significance analyzed using two-way ANOVA. (P) Naïve B cells. (Q) GC B cells.

Data information: All data show means ± SEM.

proliferation (Seligman *et al*, 1992; Cronin *et al*, 2019), we endeavored to test whether iron uptake played a role in the metabolic and survival defects in the absence of endophilin A2. Iron supplementation using ferrous sulfate or hemin chloride in both Ramos and primary B-cell cultures showed a consistent though incomplete rescue of cell growth over time (Fig 6J and L), with little effect on non-targeted or WT cells (Fig 6K and M). In contrast to iron rescue, *SH3GL1*-deleted Ramos cells were not rescued by supplementation with glucose, glutamine or non-essential amino acids (NEAA; Fig EV5A and B) and glucose uptake in these cells was normal (Fig EV5C and D). Growth defects were apparent in the context of normal mitochondrial mass in both Ramos and primary B cells (Fig EV5E and F). Furthermore, iron supplementation resulted in a partial rescue of cellular OCR in the *Sh3gl1*$^{-/-}$ B cells with little effect on WT cells (Fig 6N). As this supplementation bypasses the requirement for transferrin receptor (TfR) internalization, we analyzed transferrin uptake in CRISPR-targeted Ramos and primary naïve or GC B cells. In Ramos, *SH3GL1 targeting* reduced TfR internalization similar to PICALM depletion (Fig 6O). This reduction was recapitulated in naïve *Sh3gl1*$^{-/-}$ B cells and was particularly dramatic in GC B cells (Fig 6P and Q). Accumulation of labeled human transferrin was also significantly reduced in *SH3GL1*-deleted Ramos at both 15 and 30 min timepoints (Fig EV5G), and stronger than either PICALM or EPN1 disruption. However, unlike the BCR, TF clusters were predominantly colocalized with clathrin rather than endophilin A2 (Fig EV5H and I), pointing toward indirect involvement of endophilin A2. Together these results highlight a B cell-specific requirement for endophilin A2 in cellular metabolism, growth, and expansion, in part by regulating iron uptake.

## Discussion

Our study presents a large-scale, unbiased characterization of antigen internalization in B cells enabled by whole-genome CRISPR screening. We have validated 72 genes positively and negatively regulating BCR internalization, covering a wide range of cellular processes. As expected, the screen hits included CCP components, confirming the role of CME and pointing to a specific subset of CCP components, such as EPN1 and PICALM, that are the most intimately involved in antigen uptake. The screen also identified a number of genes regulating endosomal maturation and lysosome biology, suggesting tight coupling of BCR endocytosis to downstream trafficking pathways. Most of the identified endocytic genes are predicted to have ubiquitous roles, but we also captured proteins known to participate in BCR signaling specifically, such as CBL. The importance of BCR-specific pathways interacting with endocytic and trafficking proteins is supported by previous observations of BCR signaling inducing phosphorylation of CCP components (Stoddart *et al*, 2002) and endosomal proteins, such as the orchestrator of endosomal maturation RAB7A, one of the most highly phosphorylated proteins in B cells after antigen binding (Satpathy *et al*, 2015). Studies of B cell-specific regulation of the common endocytic proteins could expand our analysis in the future. In addition, the screen reveals many previously undescribed proteins, presenting a major leap in our understanding of this complex regulatory network.

Despite the inherent noise associated with whole-genome approaches, our data illustrate that pooled library CRISPR screening

has matured into a tool that surpasses previous approaches in efficiency and robustness. Remaining limitations for functional screens, such as ours, are gene redundancy and lethal effects of targeting genes essential for cell growth and survival. Nonetheless, our screens captured candidates that are partly redundant, such as EPN1, and genes that are also essential for growth, such as endophilin A2. The requirement of endophilin A2 for B-cell growth did prevent its identification as an endocytic regulator in our screen using the Brunello library, with subsequent analysis validating that this was due to strong depletion of *SH3GL1*-targeting sgRNAs from the cell population before the endocytic assay. Improved screening approaches in the future can further increase sensitivity of these genome-wide assays and allow more sophisticated screening for example to involve immune synapse formation. Our results show the potential of such discovery science, paving the way for other whole-genome studies in immune cells.

This work identified endophilin A2 as a novel regulator of antigen uptake in B cells. Our results are in line with the described role of endophilin A2 in FEME (Boucrot *et al*, 2015), highlighting a rapid, antigen-triggered, and clathrin-independent route of BCR endocytosis. Although our data do not completely rule out the participation of endophilin A2 in CME suggested by earlier studies (Ferguson *et al*, 2009; Milosevic *et al*, 2011; Sundborger *et al*, 2011), the distinct localization of endophilin A2 in B-cell synapses and the selective effect on antigen-induced BCR internalization contrast with CME and suggest that in B cells endophilin A2 acts predominantly in a separate, clathrin-independent pathway.

We also identified GRB2 as an important adaptor in the antigen-dependent recruitment of endophilin A2 to the BCR. Antigen ligation of the BCR leads to activation of SYK and Src-family kinases, tyrosine phosphorylation of the BCR and formation of large signaling complexes nucleated by adapter proteins including BLNK and GRB2. The molecular mechanisms by which GRB2 regulates endophilin A2 recruitment will however require further studies. GRB2 contains one SH2 and two SH3 domains, but no direct proline-rich motif to bind endophilin A2's SH3 domain, making it unlikely that it recruits endophilin A2 directly. Instead, GRB2 may stabilize recruitment of endophilin A2 through other interactions, which themselves may be redundant. Candidates from our BioID2 assays include BLNK and ITSN2 (Wong *et al*, 2012). BLNK in particular was partly required for endophilin A2 recruitment to the BCR in our experiments. ITSN2 has recently been described to be important for B-cell expansion during immune responses (Burbage *et al*, 2018b) and interacts with GRB2, providing a possible additional link between endophilin and BCR signaling complexes.

Despite regulating antigen endocytosis, endophilin A2 was dispensable for B-cell presentation of antigenic peptides on MHC II. It is possible that contribution of FEME to BCR trafficking to MHC II processing compartments is sufficiently compensated by the eventual delivery of antigen into these compartments by CME. Alternatively, FEME may actively route the BCR away from processing compartments. This is supported by our finding that the IgM BCR is degraded faster after antigen binding in endophilin A2-deficient B cells. In addition, defects in antigen processing may be compensated by the dramatically enhanced expression of MHC II on the surface of endophilin A2-deficient B cells, the basis of which remains to be investigated. Future studies of peptide processing across different compartments, as well as repertoire of presented peptides may

provide new insights. Although our data cannot currently exclude all antigen processing and presentation defects, we favor the explanation that the impairment of GC B-cell responses in endophilin A2-deficient mice is a consequence of the essential role of endophilin A2 in B cell-intrinsic growth and proliferation.

The loss of endophilin A2 also resulted in severe disruption of the MZ, an area of the spleen crucial for rapid humoral responses to blood-borne pathogens. MZ loss has been shown to result from developmental block or impaired maintenance caused by abnormal BCR, BAFFR, Notch, integrin, or chemokine receptor functions (Pillai & Cariappa, 2009). Indeed, we have detected lower IgM-BCR surface levels and reduced BAFFR and integrin expression. We could not attribute the MZ defects to a specific BCR signaling defect, and Notch target gene expression and chemokine-induced chemotaxis were also normal. It is thus possible that reduced MZ B-cell development and maintenance are associated with reduced BAFFR functions, observed in survival assays *in vitro*. Lower BAFFR function may also contribute to lower numbers of follicular B cells. Alternatively, the reduction in MZ B cells may reflect as yet unknown endosomal trafficking requirements specific for the MZ. Interestingly, dysregulation of MZ B cells observed upon loss of other endocytic regulators, such as the clathrin adaptor, EPS15 which negatively regulated antigen uptake in our screen, has been shown to lead to a significant increase in MZ B-cell numbers (Pozzi *et al*, 2012).

We have shown that endophilin A2 is crucially required for the final stage of development or maintenance of mature B cells and their response to immunological challenge. In the absence of endophilin A2, we observed reduced numbers of GC B cells, lower numbers of PCs, lower antigen-specific serum IgG1 and IgG3 antibody levels, and impaired IgG affinity maturation. This, together with normal early signaling and activation of B cells *in vitro* and *in vivo* points to a defect in B-cell expansion at the onset and during the GC reaction. Given the absence of effects on MHC II-mediated antigen presentation discussed above, the defect is best explained by the essentiality of endophilin A2 in cell-intrinsic growth of CD40-activated B cells and GC-like B-cell lymphoma cell lines. Further investigation of the defect showed that the impaired growth was associated with reduced mitochondrial respiration capacity. Supplementation with free iron partially rescued both mitochondrial respiration and cell growth in the Ramos cell line and activated primary mouse B cells. These results strongly implicate impaired iron uptake in the defect.

Iron has been increasingly linked to efficient immune defenses. Both nutritional iron deficiency and genetic defects affecting cellular iron uptake cause immunodeficiency in humans. The role of iron in the immune system has been particularly strongly linked to lymphocyte proliferation in response to antigen (Kurz *et al*, 2011; Cronin *et al*, 2019; Yambire *et al*, 2019; Weber *et al*, 2020). The uptake of iron by lymphocytes depends on endocytosis of transferrin by TfR. Our data provide novel insight into the unique regulation of iron uptake in B cells. We have observed reduced TfR endocytosis in endophilin A2-deficient B cells. These data contrast with the reported lack of effect of endophilin A2 deficiency on TfR endocytosis in T cells (Boucrot *et al*, 2015), confirmed by our results showing normal proliferation of endophilin A2-deficient T cells. However, in contrast to the BCR, surface TfR clusters colocalized predominantly with clathrin and very little with endophilin A2. Thus, endophilin A2 may have roles in iron uptake that regulate TfR endocytosis

indirectly, for example by effects on intracellular trafficking of other endocytic components.

Both the cellular growth defect and reduced TfR uptake were most prominent in the GC B-cell population. Iron homeostasis is crucial across all hematopoietic lineages (Cronin *et al*, 2019), all of which express endophilin A2 as the only endophilin A family member. The exacerbated effects in GC B cells could be related to their unique transcriptional profile, which includes changes in expression, localization, and function of endocytic and trafficking proteins (Kwak *et al*, 2018). The selective defect caused by endophilin A2 deletion in GC B cells thus points to a specialized regulation of endocytosis and iron uptake at this stage of the immune response. Such regulation may use the signaling dependency of endophilin A2's function to couple iron uptake to signals driving positive selection of GC B cells. This is consistent with the importance of endophilin A2 for affinity maturation of the antibody response.

In summary, we have shown a novel B cell-intrinsic role for endophilin A2 in antibody responses. The critical role of endophilin A2 in B-cell growth depending on trace element uptake is enhanced by cellular competition and may thus support rapidly dividing B cells in low-nutrient conditions. In light of the partial rescue of the growth defect by iron supplementation, other nutrients may become limiting as well. This unique regulation of nutrient uptake is important in protective GC responses but also in B lymphoma cells derived from the GC as indicated by published genome-wide screens, in which endophilin A2 appears to be essential selectively in cell lines derived from Burkitt and diffuse large B-cell lymphomas of the GC type. Thus, endophilin-regulated intracellular trafficking has novel implications for B-cell immunity and pathology through regulation of antigen uptake, endocytic homeostasis, and cell metabolism.

# Materials and Methods

### Mice and cell isolation

C57BL/6 and Cas9 mice on a C57BL/6 background were used as a sources of primary mouse B cells. In addition, mice were crossed with SW$_{HEL}$ (Igh$^{tm1Rbr}$-Tg(IgkHyHEL10)1Rbr) mice (Phan *et al*, 2003). To generate bone marrow chimeras, C57BL/6 mice were lethally irradiated with two doses of 5 Gy and reconstituted with bone marrow by intravenous injection (100,000 cells/ host). For *in vitro* B-cell studies, naïve murine B cells were isolated by negative selection using anti-CD43 microbeads (Miltenyi). All mice were bred and treated in accordance with guidelines set by the UK Home Office and the Francis Crick Institute Ethical Review Panel.

### Stable cell lines and lentiviral transduction

Ramos cells were maintained in full RPMI (10% fetal calf serum (FCS, BioSera), 100 μM non-essential amino acids, 20 mM HEPES, 2 mM glutamine, 50 μM beta-mercaptoethanol). DG75 cells were sourced from the Human Collection for Microorganisms and Cell Cultures in Braunschweig and maintained similarly.

To create stable Ramos Cas9 cells, recombinant replication-incompetent lentiviruses were produced by co-transfecting HEK293T cells using transIT-LT1 (Mirus) with pMD2.G and psPAX2 helper plasmids together with lentiCas9-Blast (Addgene, #52962). Lentivirus

was harvested 48 and 72 h following transfection, concentrated by ultracentrifugation, and used to spin-infect Ramos cell lines for 90 min at 1,350 *g* in the presence of 10 μg/ml polybrene (Sigma). Forty-eight hours after the spinfection, cells were selected using 10 μg/ml blasticidin. Cells were single cell sorted and expanded.

### CRISPR-mediated gene disruption

CRISPR sgRNA sequences were designed using the Broad Institute's sgRNA Designer. Forward and reverse oligonucleotides including the guide sequence were synthesized, phosphorylated, annealed, and individually cloned into lentiGuide-Puro or lentiGuide-mCherry (in which puromycin resistance cassette was replaced by mCherry) plasmids. Lentivirus was produced and cells spinfected as described above. Cells were selected using 2.5 μg/ml of puromycin. CRISPR-targeted cells were used within two weeks of selection. Cells expressing a fluorescent marker were further purified using Avalon cell sorter (Propel Labs).

### CRISPR screens and analysis

GeCKO and Brunello whole-genome human libraries were acquired from Addgene. Smaller scale validation library was created by selecting the top 300 positive and 50 negative regulators from the Brunello screen and filtering for expression in Ramos cells; non-targeting and essential gene controls were added to the 213 experimental genes.

Stable Ramos Cas9 cells were expanded and $80 \times 10^6$ cells spinfected with pooled lentiviral whole-genome CRISPR libraries at multiplicity of infection of 0.5, achieving 500× library coverage. This was appropriately scaled down for custom library infections. A sample of the infected cells was removed for genomic DNA analysis; the rest were selected with 2.5 μg/ml puromycin for 3 weeks and maintained at 1,000× library coverage at every passage. Internalization assays were performed as described below. Sorted "internalized" and "non-internalized" populations, as well as whole populations pre- and post-selection, were used to extract gDNA (DNeasy Blood & Tissue kit, Qiagen) and amplify the integrated viral genome using a nested PCR, as described in Shalem *et al* (2014). Libraries were sequenced on a MiSeq system (Illumina). Raw reads in demultiplexed FASTQ files were trimmed to contain sgRNA sequences only, and these were aligned to the original library using Bowtie. The number of aligned reads for each unique sgRNA was counted and used to calculate survival and internalization scores. Analysis was performed in MATLAB (MathWorks) and statistics were calculated using MAGeCK (Li *et al*, 2014).

Internalization score

$$= \log_2 \left( \frac{1 + \text{sgRNA abundance non} - \text{internalized population}}{1 + \text{sgRNA abundance internalized population}} \right).$$

### Surrogate antigens

Model antigens used in solution or on PMS were goat anti-mouse Igκ F(ab')$_2$ (Southern Biotech) for mouse splenic B cells and goat F(ab')$_2$ anti-human Fc5μ (Jackson Immunoresearch) for Ramos and DG75 cells. The antibodies were biotinylated using EZ-Link NHS-LC-LC-biotin (Pierce) and conjugated to one of several fluorophores - Cy5

Monoreactive dye (GE Healthcare), AlexaFluor 405 or AlexaFluor 647 NHS esters (Thermo Fisher) in Sodium Carbonate buffer, according to the manufacturer's instructions. Excess dye was removed using Zeba 7K MWCO desalting columns (Pierce, Thermo Fisher). Mouse transferrin (Rockland) was also fluorescently labeled and biotinylated for TfR internalization assays.

### Soluble antigen internalization assay

Murine B cells or Ramos cells were stained with a LIVE/DEAD marker then washed and incubated with the surrogate antigens described above at 2 μg/ml in PBS- BSA (1%) for 10 min on ice. Cells were washed and incubated on ice or at 37°C for a set time frame (5–30 min) before fixation in 4% PFA. Washed cells were then stained for 30 min on ice with fluorescently labeled streptavidin, to counterstain remaining surface antigen. Samples were analyzed on a BD LSRFortessa. Internalization was quantified as the percentage of antigen remaining at the cell surface (streptavidin-labeled) using FlowJo (TreeStar). For large-scale screening assays, a minimum of 120 million cells was processed as described and highest and lowest internalizing populations (top and bottom 10–15%) were sorted on a BD FACSAria.

### Live and fixed cell imaging

Epifluorescence and TIRF imaging were carried out on a Nikon Eclipse Ti microscope with an ORCA-Flash 4.0 V3 digital complementary metal-oxide semiconductor (CMOS) camera (Hamamatsu Photonics) and 100× TIRF objective (Nikon). Illumination was supplied by 405, 488, 552, and 637 nm lasers (Cairn) through an iLas$^2$ Targeted Laser Illuminator (Gataca Systems) which produces a 360° spinning beam with adjustable TIRF illumination angle.

Plasma membrane sheets (PMS) were generated as described previously (Nowosad & Tolar, 2017) in 8-well Lab-Tek imaging chambers (Thermo). Isolated primary B cells or Ramos cells were washed and added to pre-warmed imaging wells, allowed to interact with antigen-loaded PMS at 37°C. Cells were imaged live or fixed in 4% PFA after 20 min.

Acquired datasets were analyzed using automated routines in ImageJ and MATLAB (MathWorks). All cells imaged were included in the analyses. Substrate-bound antigen internalization was quantified by detecting antigen clusters extracted from the PMSs inside the B cells identified using B220 or CD19 surface staining as described previously (Nowosad *et al*, 2016). Endophilin A2-GFP and clathrin-mCherry spots were detected as described (Roper *et al*, 2019). Briefly, Endophilin A2 or clathrin TIRF images were enhanced by convolution with a 325 nm-sized bandpass filter and spots above threshold were tracked and analyzed for fluorescence intensities in all relevant channels. Thresholds were set automatically as a fraction of each cell's mean fluorescence intensity. To analyzed Endophilin A2-clathrin colocalization, coinciding spots were detected as those whose centers were less than 128 nm apart. Random spot colocalization was calculated after randomly picking same numbers of non-overlapping spots from each cell's area.

### Antibodies and flow cytometry

Erythrocyte-lysed single-cell suspensions were blocked with anti-CD16/32 for 15 min and stained with appropriate antibodies for

30 min on ice. The following stains and antibodies were used for immunophenotyping: B220 (RA3-6B2), CD25 (M-A251), CD69 (H1.3F2), CD86 (GL1), and CD95 (Jo2) from BD Bioscience; CD19 (1D3), CD23 (B3B4), CD38 (90), CD93 (AA4.1), IgM (II/41), and MHCII (M5/114.15.2) from eBioscience; and CD21/CD35 (7E9), CD45.1 (A20), CD138 (281-2), and c-kit (2B8) from BioLegend.

In spleen, follicular and marginal zone cells were detected based on CD23 and CD21/35 expression in the mature cell population (B220$^+$CD93$^-$). Transitional populations were detected in the immature population (B220$^+$CD93$^+$) based on expression of CD23 and IgM. Germinal center cells were defined as B220$^+$ GL7$^+$Fas$^+$CD38$^-$ population; plasma cells were detected by expression of CD138. In the bone marrow, early B-cell populations were defined as follows: pre-pro-B (B200$^+$c-kit$^+$CD25$^-$CD19$^-$), pro-B (B200$^+$c-kit$^+$CD25$^-$CD19$^+$), and pre-B cells (B200$^+$c-kit$^-$CD25$^+$CD19$^+$).

## Western blot, immunoprecipitation, and IgM degradation assay

For signaling assays, a minimum of $5 \times 10^6$ cells were activated with soluble model antigens at 37°C for set time points, then washed, and processed on ice. Cells were lysed in RIPA buffer (Sigma), containing cOmplete EDTA-free protease inhibitor cocktail (Roche), for 10 min on ice and then centrifuged at 16,000 g for 10 min at 4°C. Samples were boiled in 1X NuPAGE LDS Sample Buffer (Invitrogen) and 1X NuPAGE Sample Reducing Buffer (Invitrogen) at 95°C for 5 min and then separated using ExpressPlus PAGE Gels 4–20% (GenScript). Proteins were transferred to a polyvinylidene fluoride (PVDF) membrane and blocked for 1 h in PBST with 50% Odyssey Blocking Buffer (LI-COR). Primary antibodies were incubated overnight in the same blocking solution; fluorescently tagged secondary antibodies were incubated for 1 h. Membranes were imaged using an Odyssey CLx or Odyssey Fc imaging systems (LI-COR).

For immunoprecipitation, cell lysates were incubated with biotinylated pull-down antibodies and streptavidin-M280 Dynabeads (Invitrogen) overnight at 4°C. Beads were washed on a Dynabead magnet 5 times and eluted in 1× NuPAGE Sample Reducing Buffer at 95°C. Samples were processed as for Western blot above.

For IgM degradation assays, $30 \times 10^6$ primary mouse cells were surface biotinylated using EZ-Link Sulfo-NHS-LC-LC-biotin (Pierce) as per manufacturer's instructions. After labeling, cells were washed in PBS with 100 mM glycine to quench and remove excess biotin. Cells were washed three times and incubated with anti-IgM on ice for 15 min. After washing, cells were incubated at 37°C for 0, 3, 6, or 12 h to allow internalization and recycling or degradation of surface proteins. Cells were then lysed; biotinylated proteins were pulled down using streptavidin-M280 Dynabeads and both supernatants and eluates processed for Western blot.

## Gene expression analysis

RNA was isolated from sorted follicular B cells, ex vivo or after overnight CD40L culture, using RNeasy mini kit (Qiagen). Poly-A enriched libraries were prepared using SMART-Seq HT and NexteraXT (Illumina) and sequenced on Illumina HiSeq 2500. Reads were aligned to C57BL/6J mouse reference genome mm10, GRCm38 and quantified using Tophet and Cufflinks.

## BrdU pulse-chase

CRISPR-targeted Ramos cells were loaded with BrdU (Sigma) at final concentration of 10 µM in full RPMI at 37°C for 1 h. Cells were washed and cultured at 37°C. They were harvested and processed at various time points to follow the labeled cohort through the cell cycle. Fixation was performed in cold 70% ethanol for 30 min with regulator vortexing. Samples were washed in PBS+ 2% BSA, resuspended in 500 µl of 2 M hydrochloric acid, and incubated for 20 min. Samples were washed three times to remove acid traces and labeled sequentially with anti-BrdU antibody (Becton Dickinson) at 1/50 dilution in PBT-tween for 30 min; goat anti-mouse secondary antibody (AF488, Invitrogen) for 20 min; and anti-MPM2-Cy5 antibody (EMD Millipore) at 1/200 dilution for 1 h, with thorough washing between stains. Finally, samples were stained with propidium iodide and analyzed by flow cytometry. Mitotic cells were detected as MPM2-positive; G1, S, and G2 were identified by DNA content (PI); BrdU$^+$ proportion was quantified within each population at given timepoints.

## Antigen presentation assays

To detect antigen presentation in vitro, B cells were incubated with microbeads (Bangs laboratories) conjugated to anti-Igκ and Ea peptide (Biotin-GSGFAKFASFEAQGALANIAVDKA-COOH) for 5 h at 37°C, fixed in 4% PFA, and blocked with 2% BSA and Fc block.

For in vivo assays, mice were immunized (intraperitoneal) with SRBC-HEL conjugated to Ea peptide, as previously described (Brink et al, 2015). Splenocytes were harvested 24 h later and blocked with 2% BSA and Fc block on ice. For both assays, samples were then stained with antibody against Ea52–68 bound to I-Ab (eBioY-ae, eBioscience) at 1/100 dilution in PBS-BSA.

## Cytokine cultures and iron supplementation

To study proliferation, B cells were labeled with 1 µM DDAO (Thermo) and cultured in full RPMI supplemented with 4 µg/ml CD40L (R&D systems), 100 ng/ml BAFF (PeproTech), 1 µg/ml anti-IgM (Jackson Immunoresearch), or 2.5 µg/ml CpG (Invitrogen). DDAO dilution (and mCherry expression in CRISPR-targeted B cells) was detected by flow cytometry 3 days post-culture.

For cell expansion studies, full RPMI was supplemented with one of the following: 5 µM hemin chloride (Sigma), 50 µg/ml ferrous sulfate (Sigma), 10 g/l D-Glucose (Gibco), 10 mM L-Glutamine (Gibco), or 5× non-essential amino acids (NEAA, Gibco).

## Calcium

Cells were loaded with Fluo-4 and Fura-Red as per manufacturer's instructions for 30–45 min at 37°C in RPMI$^+$ 5% FCS. Cells were washed and analyzed by flow cytometry in HBSS-0.1% BSA. At 20s of acquisition, CRISPR-targeted Ramos were activated with 2 µg/ml anti-human IgM F(ab')$^2$; primary B cells from Sh3gl1$^{-/-}$ mice or WT littermates were activated with 2 µg/ml anti-mouse Igκ F(ab')$^2$. Samples were acquired for 380 s, with addition of ionomycin (5 µg/ml) at 300 s.

## Migration assay

Directional migration assays were performed using 96-well Transwell plates with 5 μm polycarbonate membranes (Sigma-Aldrich). Top compartments were coated with 0.5 μg/ml ICAM-1 or 1μg/ml anti-Igκ F(ab')$_2$ in PBS for 1 h, following by 20 min blocking in PBS-BSA (1%). Bottom compartments contained RPMI with 1 μg/ml CXCL13, 0.5 μg/ml CXCL12, or 1 μg/ml CCL21 (all from R&D systems). Isolated chimera B cells, of known mCherry infection rates, were added to the top compartments and allowed to migrate at 37°C for 3 h. mCherry percentages were quantified by flow cytometry.

## BioID2 proximity assay

SH3GL1-BioID2-HA and empty-BioID2-HA constructs were cloned into a lentiviral expression vector (pLenti-Puro, Addgene) and used to create single-cell cloned stable Ramos cell lines. Expression was tested by Western blot and correct localization confirmed by immunofluorescence using anti-BioID2 antibody (SS GD1, Novus Biologicals). $1 \times 10^8$ cells per sample were cultured in 1 μM biotin in full RPMI for 14 h. Cells were washed and lysed in RIPA buffer containing protease inhibitor cocktail. Protein concentration was determined using a BCA assay (Sigma) and biotinylated proteins were pulled down using a corresponding concentration of Dynabeads Streptavidin C1 (Thermo), as per manufacturer's instructions. Biotinylated fraction was eluted in NuPAGE Sample Reducing Buffer (SDS-containing; Thermo), boiled at 95°C for 5 min and in-gel digestion performed. Peptide extracts were resolved on an EASY-Spray column (Thermo) using UltiMate 3000 RSLCnano System. Data acquisition was performed using an Orbitrap mass spectrometer (Thermo) controlled by Xcalibur software. Data analysis was performed using the MaxQuant bioinformatics suite, using LFQ (Label Free Quantification) algorithm.

## Seahorse assay

Agilent Seahorse XF assay was performed as per manufacturer's instructions. Briefly, B cells were seeded in poly-L-lysine-coated Agilent culture plate at 500,000/well for primary cells and 250,000 for Ramos B cells. Cells were cultured in XF Base Medium supplemented with pyruvate, glutamine, and glucose for 1 h in a non-CO$_2$ incubator at 37°C. Stress compounds (all from Sigma) were diluted in assay medium and loaded at 10× concentration into the injection ports of the hydrated cartridge as follows: A—Oligomycin, 1 μM final concentration; B—FCCP (Carbonyl cyanide-*p*-trifluoromethoxyphenyl hydrazone), 1.5 μM final; C—Rotenone, 200 nM final. XF Cell Mito Stress Test was run on a Seahorse XF 96 Analyzer with Wave software. Data were analyzed using Wave and GraphPad Prism.

## Immunization and serum antibody ELISA

*Sh3gl1*$^{-/-}$ and WT littermates were immunized at 8–10 weeks of age by intraperitoneal injection of 50 μg NP-CGG in alum (Thermo Fisher). Blood samples were taken before immunization and at 7 and 14 days after immunization. Serum NP-specific antibodies were detected by ELISA on Nunc polysorb plates coated with NP$_7$ or NP$_{25}$ for capture. Total serum immunoglobulin levels were detected using SBA Clonotyping System HRP kit (Southern Biotech).

## Statistics

For *in vivo* experiments, sex and age matched mice were randomly assigned to groups and housed in a randomized manner. Animal code rather than genotype was used throughout to avoid bias. Investigator blinding was used in ELISA and antigen presentation experiments. Sample sizes for individual experiments are detailed in figure legends. Appropriate statistical analysis was performed in GraphPad Prism 9.

# Data availability

Datasets produced in this study are available in the following databases:
- RNA-Seq data: Gene Expression Omnibus GSE172295 (https://www.ncbi.nlm.nih.gov/geo/query/acc.cgi?acc=GSE172295).
- BioID2 protein interactions: EMBL-EBI BioStudies S-BSST665 (https://www.ebi.ac.uk/biostudies/studies/S-BSST665).

**Expanded View** for this article is available online.

## Acknowledgements

This work was supported by the European Research Council (Consolidator Grant 648228) and the Francis Crick Institute, which receives its core funding from Cancer Research UK (FC001185), the UK Medical Research Council (FC001185), and the Wellcome Trust (FC001185). This research was funded in whole, or in part, by the Wellcome Trust (Grant number FC001185). For the purpose of Open Access, the author has applied a CC BY public copyright license to any Author Accepted Manuscript version arising from this submission. We thank Emmanuel Boucrot and Jürgen Wienands for helpful discussions and the Francis Crick Institute Science Technology Platforms (STPs) for their help with mass spectrometry, cell sorting, and next-generation sequencing and Harshil Patel (Bioinformatics and Biostatistics STP) for RNAseq data analysis.

## Author contributions

Designing and performing experiments and data analysis: DM; live cell imaging: LW; GC antigen presentation assay: RN and AM-R; knock out DG75 cells and advice: NE; experiment design, data analysis, and research supervision: PT; manuscript preparation: DM and PT.

## Conflict of interest

The authors declare that they have no conflict of interest.

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
