## [Review Process File · EMBO Reports]

Endophilin A2 regulates B cell endocytosis and is required for germinal center and humoral responses

Dessislava Malinova, Laabiah Wasim, Rebecca Newman, Ana Martinez-Riaño, Niklas Engels, and Pavel Tolar

DOI: [10.15252/embr.202051328](https://doi.org/10.15252/embr.202051328)

Corresponding author(s): Pavel Tolar (pavel.tolar@crick.ac.uk) , Dessislava Malinova (d.malinova@qub.ac.uk)

Review Timeline:

Submission Date:	14th Jul 20
Editorial Decision:	24th Aug 20
Revision Received:	30th Mar 21
Editorial Decision:	15th Jun 21
Revision Received:	22nd Jun 21
Accepted:	9th Jul 21

Transaction Report:

Dear Dr. Tolar

Thank you for the submission of your research manuscript to our journal. We have now received the full set of referee reports that is copied below.

As you will see, the referees acknowledge that the findings are potentially interesting. However, they also point out several technical concerns and have a number of suggestions for how the study should be strengthened, and I think that all of them should be addressed. In particular, the specificity of Endophilin A2 function to germinal center B cells and to BCR endocytosis should be revisited and the data on its clathrin-independent role should be strengthened.

Given these constructive comments, we would like to invite you to revise your manuscript with the understanding that the referee concerns (as detailed above and in their reports) must be fully addressed and their suggestions taken on board. Please address all referee concerns in a complete point-by-point response. Acceptance of the manuscript will depend on a positive outcome of a second round of review. It is EMBO reports policy to allow a single round of revision only and acceptance or rejection of the manuscript will therefore depend on the completeness of your responses included in the next, final version of the manuscript.

We invite you to submit your manuscript within three months of a request for revision. This would be November 24th in your case. Yet, given the current COVID-19 related lockdowns of laboratories, we have extended the revision time for all research manuscripts under our scooping protection to allow for the extra time required to address essential experimental issues. Please contact us to discuss the time needed and the revisions further.

- 1) A data availability section is missing.
- 2) Your manuscript contains error bars based on $n=2$. Please use scatter blots showing the individual datapoints in these cases. The use of statistical tests needs to be justified.

Please note that for all articles published beginning 1 July 2020, the EMBO Reports reference style will change to the Harvard style for all article types. Details and examples are provided at <https://www.embopress.org/page/journal/14693178/authorguide#referencesformat>

- 1) a .docx formatted version of the manuscript text (including legends for main figures, EV figures and tables). Please make sure that the changes are highlighted to be clearly visible.
- 2) individual production quality figure files as .eps, .tif, .jpg (one file per figure). Please download our Figure Preparation Guidelines (figure preparation pdf) from our Author Guidelines pages

<https://www.embopress.org/page/journal/14693178/authorguide> for more info on how to prepare your figures.

4) a complete author checklist, which you can download from our author guidelines (). Please insert information in the checklist that is also reflected in the manuscript. The completed author checklist will also be part of the RPF.

5) Please note that all corresponding authors are required to supply an ORCID ID for their name upon submission of a revised manuscript (). Please find instructions on how to link your ORCID ID to your account in our manuscript tracking system in our Author guidelines ()

6) We replaced Supplementary Information with Expanded View (EV) Figures and Tables that are collapsible/expandable online. A maximum of 5 EV Figures can be typeset. EV Figures should be cited as 'Figure EV1, Figure EV2" etc... in the text and their respective legends should be included in the main text after the legends of regular figures.

- For the figures that you do NOT wish to display as Expanded View figures, they should be bundled together with their legends in a single PDF file called *Appendix*, which should start with a short Table of Content (incl. page numbers). Appendix figures should be referred to in the main text as: "Appendix Figure S1, Appendix Figure S2" etc. See detailed instructions regarding expanded view here:

- Supplementary tables S1-S5 are complex tables and should be uploaded as Dataset (Dataset EV1, etc). The legends for these have to be provided in a separate tab of the .xls files. Alternatively, the legend can be supplied as a separate text file (README) and zipped together with the Table/Dataset file.

- Movies: please provide the legend as separate text file (README) and zip it together with the movie. Please note the nomenclature Movie EV1.

7) Please note that a Data Availability section at the end of Materials and Methods is now mandatory. In case you have no data that requires deposition in a public database, please state so instead of refereeing to the database.

See also < <https://www.embopress.org/page/journal/14693178/authorguide#dataavailability>>).

Please note that the Data Availability Section is restricted to new primary data that are part of this study.

8) We would also encourage you to include the source data for figure panels that show essential data. Numerical data should be provided as individual .xls or .csv files (including a tab describing the data). For blots or microscopy, uncropped images should be submitted (using a zip archive if multiple images need to be supplied for one panel). Additional information on source data and instruction on how to label the files are available .

10) Regarding data quantification:

- Please ensure to specify the name of the statistical test used to generate error bars and P values, the number (n) of independent experiments (please specify technical or biological replicates) underlying each data point and the test used to calculate p-values in each figure legend. Discussion of statistical methodology can be reported in the materials and methods section, but figure legends should contain a basic description of n, P and the test applied.
- Graphs must include a description of the bars and the error bars (s.d., s.e.m.).
- Please also include scale bars in all microscopy images.

11) As part of the EMBO publication's Transparent Editorial Process, EMBO reports publishes online a Review Process File to accompany accepted manuscripts. This File will be published in conjunction with your paper and will include the referee reports, your point-by-point response and all pertinent correspondence relating to the manuscript.

I look forward to seeing a revised version of your manuscript when it is ready. Please let me know if you have questions or comments regarding the revision.

Yours sincerely

Martina Rembold, PhD
Editor
EMBO reports

Referee #1:

In this study, by using systematic screening, authors have identified endophilinA2 for one of important components for antigen-internalization and tried to clarify its biological functions in B cell biology. I think that this is a nice challenging study, but that the observations are fragmentary and

their interpretations are sometimes not well considered.

- 1) In Fig. 2, authors should examine the effects of clathrin sg, because one of the important conclusions in this manuscript is that endophilinA2 mediates in a clathrin-independent manner.
- 2) In Fig. 3D, CD86 expression is apparently decreased in Sh3gl1-targeted B cells. But, authors mentioned similarly. Given the biological importance of CD86, authors should carefully interpret, otherwise the conclusion might be wrong.
- 3) In Fig. 3F, both recruitment of B cells into GC and maintenance of GC B cells appear to be affected in Sh3gl1-targeted B cells. This point should be more carefully examined. And, authors did not examine the GC status, for example, LZ/DZ and proliferation/survival.
- 4) In Fig. 4, expression of ClassII, BAFFR, LPAM1 was so much affected in endophilinA2 ko B cells. Furthermore, authors mentioned lower surface BCR and integrin expression in the discussion part, although I could not find out these data. Particularly, expression level of BCR is very important; authors should show up. Moreover, given the biological importance of these cell surface molecules, authors should consider seriously why such differences come up by ko B cells and what are the biological consequences.
- 5) Authors examine the numbers of Tfh cells in Fig. 4H, which is very nice. But authors should examine the function of the ko Tfh cells. Thus, the current T cell data shown by authors do not necessarily indicate no T cell anomalies in endophilinA2 ko mice. To minimize the effects by T cell anomalies, authors are encouraged to use mixed bone marrow chimera using μ M ko.
- 6) Figure 6 data are nice, but as authors point out, these are more downstream events. It is important to connect these findings to endophilinA2 function.

Referee #2:

This is an interesting and novel report on a global antigen internalization study of B cells using a high throughput CRISPR screen. To my knowledge, such a study has not been done before. The authors work with a B cell line and use two independent guide RNA libraries and identify a couple of proteins involved in endocytosis. Of these proteins they follow up one identified positive regulator called endophilin A2. They follow up the function of this protein and show that it is not crucial for antigen presentation of B cells to T cells, but plays a role in germinal centre B cells, affecting proliferation, metabolism and immune responses by these cells. For the latter studies they use endophilin A2 KO mice. This is a comprehensive and well performed study which also determines a mechanism to a great extent. It should be published. I have only some minor points to be addressed.

- 1) The authors throughout the study use anti-IgM Fab2 and call this antigen. It is a stimulating antibody fragment, not binding to the antigen-binding side of the BCR, therefore there is some discussion how well it replaces natural antigen. I would only call it surrogate antigen at best. It should always be mentioned that it is an antibody fragment instead.
- 2) In Fig.1 D the authors use a mini library validation approach. Why is SH3GL1 not included? Or was it included and not significant? Please comment and mention in the results section.
- 3) Fig.2G, please also show the colocalization of endophilin A2 and "antigen" as a comparison, as this is relevant.
- 4) Fig. 2 H the label on the blots "a-IgM pulldown" is confusing, as the band shows endophilin. I would label it as "immunoprecipitated endophilin"
- 5) Fig. 5E: Downregulation of IgM. On which cells was this gated? Total B cells? As the MZ B cells are high on IgM this could affect the analysis. Should be gated on follicular B cells.

Referee #3:

This manuscript presents a large amount of data. It describes a research story starting with identifying new components in B-cell receptor (BCR) endocytosis and ending by searching the roles of endophilin A2 in BCR-independent survival and proliferation of peripheral B-cells. This study is the first to use CRISPR-based screening to identify genes regulating BCR-mediated antigen internalization, a critical step in the activation of humoral immune responses. Although the data reveal weaknesses in the cellular assays for the screen, this manuscript presents a new application for the CRISPR-based screening in the field. The screen identified endophilin A2, a component of clathrin-independent endocytosis pathways. The authors developed a human B-cell line, bone marrow chimeric mice, and knockout mice and confirmed the involvement of endophilin A2 in BCR-mediated antigen internalization. However, endophilin A2-dependent BCR internalization did not significantly contribute to BCR-mediated antigen processing and presentation. Instead, endophilin A2 inhibits the turnover and maintains the surface level of the BCR. Therefore, endophilin A2-dependent BCR endocytosis cannot explain the immunological phenotypes of endophilin A2-deficient mice - decreases in follicular and germinal center B cells and IgG levels. After finding that endophilin A2-deficiency inhibits B-cell proliferation in response to a wide arrange of stimuli, the authors turned to examine B-cell metabolism, specifically oxygen consumption rates. The reduction in the oxygen consumption rates leads to the discovery of the defect in iron uptake in endophilin A2-deficient B-cells, including transferrin receptor endocytosis. This study utilized a banquet of cutting edge technologies and generated new information on the cell biology of B-cells, which should be the interest of the readers of EMBO Reports. However, the manuscript lost its focus on endocytosis in the middle and has not got deep into the mechanism by which endophilin A2-mediated endocytosis uniquely regulates B-cell function.

Specific comments:

1. The title is not precise by starting with germinal center B-cell responses. The data presented in the manuscript show that the involvement of endophilin A2 is not exclusive to germinal center B-cells. For example, the number of follicular B-cells was also reduced in endophilin A2-deficient mice. IgG reduction could be due to defects in germinal center B-cells or non-germinal center B-cells, as isotype switching does occur outside germinal centers. The manuscript does not provide any data that compare BCR and transferrin endocytosis in germinal center B-cells with other B-cell subsets. Even though mice were immunized, the majority of their B-cells are not germinal center B-cells.
2. The BCR endocytosis assay used for screening may not be appropriately performed. This flow cytometry-based BCR endocytosis assay is well established in the literature. The manuscript only shows two small dot plots in Figure 1A. Based on these two plots, the fluorescence of the streptavidin staining (surface BCRs) and the anti-BCR staining (total BCRs) significantly overlap with each other. Such overlap may not ruin the results but dramatically reduced the sensitivity of the screening.
3. Endophilin A2 has been shown to contribute to the endocytosis and intracellular trafficking of several receptors in various cell types. After the initial screening with the BCR endocytosis assay, additional assays should be applied to determine if the involvement of endophilin A2 in endocytosis is exclusive to the BCR. Transferrin receptor endocytosis is commonly used for this purpose. If transferrin receptor endocytosis is affected by endophilin A2-deficiency, the endocytosis of additional receptors should be tested to estimate the involvement scope of endophilin A2-dependent endocytosis in B-cells. Under this scenario, the oxygen consumption rate analysis becomes unnecessary.

4. The antigen internalization versus time plots in Figure 2C and 5A-C were not presented correctly. The time scale is not scientific, which gives a false impression of endocytosis kinetics. If the authors like to link the data points by lines, 5 to 15 min in the x-axis should be three times the length of 0 to 5 min.

5. Figure 2 should include the percentages of antigen dots colocalizing with endophilin A2 and clathrin using TIRF images. Such data reflect the relative fractions of BCRs that undergo endophilin A2- or clathrin-mediated endocytosis. There are much more of endophilin A2 dots than antigen dots in TIRF images. This brings to the next point: the authors should make sure that the overexpression of fluorescent fusion proteins did not artificially affect B-cells and confirm the critical results by detecting the endogenous endophilin A2, including the pull-down assay in Figure 2H.

6. Figure 3F should include follicular B-cells.

7. The effect of endophilin A2-deficiency on the turnover rate of membrane IgM is very interesting, which may be one of the B-cell specific roles of endophilin A2. Does endophilin A2-deficiency has a similar effect on mIgG turnover? The western blots showed in Figure 5F should be labeled with membrane IgM (mIgM) rather than IgM. A molecular weight marker should be added.

8. The data transferrin endocytosis in Figure S7G should be included in Figure 6. This is the only data in this manuscript linking endophilin A2-mediated endocytosis to B-cell survival and proliferation.

Response to reviewers

Referee #1:

In this study, by using systematic screening, authors have identified endophilinA2 for one of important components for antigen-internalization and tried to clarify its biological functions in B cell biology. I think that this is a nice challenging study, but that the observations are fragmentary and their interpretations are sometimes not well considered.

1)In Fig. 2, authors should examine the effects of clathrin sg, because one of the important conclusions in this manuscript is that endophilinA2 mediates in a clathrin-independent manner.

We found that analysis of clathrin-targeted cells is difficult because of poor viability of these cells. We have therefore quantified localization of endophilin A2 in PICALM and EPN1 knock out cells, which do not have a growth defect, but do have a defect in clathrin-mediated BCR endocytosis. In these cells, the colocalization of clusters of surrogate antigen with endophilin A2-GFP was comparable to wildtype (Fig 2I). These new results, together with new Fig 2H showing independent colocalization of surrogate antigen clusters with endophilin A2 and clathrin spots (Fig 2H) strongly support our data that endophilin A2 contributes to BCR endocytosis independently of clathrin coated pits. These latter results have also been confirmed using staining for endogenous endophilin A2 and clathrin (see response to Referee 3).

2)In Fig. 3D, CD86 expression is apparently decreased in Sh3gl1-targeted B cells. But, authors mentioned similarly. Given the biological importance of CD86, authors should carefully interpret, otherwise the conclusion might be wrong.

The CD86 fluorescence intensity is quantified in Fig 3E. It is slightly lower in endophilin A2 knockout cells, but the difference is not statistically significant ($p=0.3205$ using 2-way ANOVA). Given that the *in vivo* phenotype is B cell intrinsic, and that there is a B cell growth defect *in vitro* in the absence of T cells, we conclude that the reduced CD86 expression is not a major factor in the Sh3gl1-KO phenotype.

3)In Fig. 3F, both recruitment of B cells into GC and maintenance of GC B cells appear to be affected in Sh3gl1-targeted B cells. This point should be more carefully examined. And, authors did not examine the GC status, for example, LZ/DZ and proliferation/survival. Indeed, loss of endophilin A2 appears to affect follicular B cell numbers and the numbers of activated B cells from the pre-germinal centre (GC) activation point onwards. The GC and plasma cell populations are the most strongly affected. Given that *in vitro*, both survival and proliferation are reduced, and these defects are further exacerbated by competition, it is likely that the all GC B cells are affected from an early time point. We have edited the manuscript and the title to reflect the comprehensive nature of the defect in multiple B cell populations.

4)In Fig. 4, expression of ClassII, BAFFR, LPAM1 was so much affected in endophilinA2 ko B cells. Furthermore, authors mentioned lower surface BCR and integrin expression in the discussion part, although I could not find out these data. Particularly, expression level of BCR is very important; authors should show up. Moreover, given the biological importance of these cell surface molecules, authors should consider seriously why such differences come up by ko B cells and what are the biological consequences.

The surface expression of the BCR is quantified in Fig 5E, showing reduced surface IgM and no difference in IgD compared to wildtype B cells. Fig 5F and G also quantify surface IgM (time = 0) and IgM degradation over time following activation. Despite the reduced surface IgM levels, we found no significant differences in early BCR signalling in endophilin A2 knockout cells, suggesting reduced BCR is unlikely to explain the phenotype.

The integrin quantified was LPAM1 (alpha4-beta7 integrin; Fig 4C) due to its role in cell migration and homing. However, transwell migration experiments (Fig EV 5C) show normal cell motility in response to a range of chemokines and adhesion molecules, suggesting that endophilin A2 is not essential for B cell adhesion and migration, although future studies are needed to examine possible subtle effects.

Surface MHCII levels were found to be significantly increased in endophilin A2 knockout B cells and this may compensate for any decrease in antigen processing. Again, future studies will be required to understand the mechanism by which endophilin A2 regulates MHCII regulation.

5) Authors examine the numbers of Tfh cells in Fig. 4H, which is very nice. But authors should examine the function of the ko Tfh cells. Thus, the current T cell data shown by authors do not necessarily indicate no T cell anomalies in endophilinA2 ko mice. To minimize the effects by T cell anomalies, authors are encouraged to use mixed bone marrow chimera using μ M ko.

We have not directly addressed Tfh function, because its potential defects are unlikely to contribute to the GC B cells phenotype, which is B cell intrinsic (Sh3gl1-targeted cells represent only ~ 50% of the haematopoietic cells in the mice, similar to mixed bone marrow chimeras). In general, endophilin A2 knockout did not have appreciable effects on CD4, CD8 and Tfh cell numbers (Fig 4), T cell proliferation (Fig 5), metabolism (Fig 6) or transcriptional status (RNAseq, not shown). Although it is possible that endophilin A2 has a role in some functions of Tfh cells we believe that these will be best left for dedicated studies.

6) Figure 6 data are nice, but as authors point out, these are more downstream events. It is important to connect these findings to endophilinA2 function.

The metabolic differences appear to result in part from reduced internalisation of transferrin receptor in Ramos and primary cells (Fig EV7). We have added data to show transferrin, and therefore iron, uptake is significantly reduced upon endophilin A2 deletion in Ramos (Fig EV7 J), linking endophilin A2-mediated trafficking with the metabolism defects rescued by iron supplementation. However, unlike the BCR, which showed strong colocalization with endophilin A2 and interaction detected by co-immunoprecipitation, the effect of endophilin A2 on transferrin internalisation appears to be indirect as transferrin clusters primarily colocalise with clathrin and only very modestly with endophilin A2 (Fig EV7 K, L). The mechanisms by which endophilin A2 regulates transferrin uptake thus needs further research but may resemble the complex effects endophilin absence has on neuronal cells.

Referee #2:

This is an interesting and novel report on a global antigen internalization study of B cells using a high throughput CRISPR screen. To my knowledge, such a study has not been done before. The authors work with a B cell line and use two independent guide RNA libraries and identify a couple of proteins involved in endocytosis. Of these proteins they follow up one identified positive regulator called endophilin A2. They follow up the function of this protein and show that it is not crucial for antigen presentation of B cells to T cells, but plays a role in germinal centre B cells, affecting proliferation, metabolism and immune responses by these cells. For the latter studies they use endophilin A2 KO mice. This is a comprehensive and well performed study which also determines a mechanism to a great extent. It should be published. I have only some minor points to be addressed.

1) The authors throughout the study use anti-IgM Fab2 and call this antigen. It is a stimulating antibody fragment, not binding to the antigen-binding side of the BCR, therefore there is some discussion how well it replaces natural antigen. I would only call it surrogate antigen at best. It should always be mentioned that it is an antibody fragment instead.

Thank you for pointing this out. We corrected the nomenclature in the text.

2) In Fig.1 D the authors use a mini library validation approach. Why is SH3GL1 not included? Or was it included and not significant? Please comment and mention in the results section.

The validation mini library was based on the results from the Brunello library. We did not include SH3GL1 in the mini library because we have already validated it manually.

3) Fig.2G, please also show the colocalization of endophilin A2 and "antigen" as a comparison, as this is relevant.

We agree and have added quantification of surrogate antigen cluster colocalization with either CME or FEME. This new analysis is now shown in Fig 2H.

4) Fig. 2 H the label on the blots "a-IgM pulldown " is confusing, as the band shows endophilin. I would label it as "immunoprecipitated endophilin"

This has been corrected in the figure.

5) Fig. 5E: Downregulation of IgM. On which cells was this gated? Total B cells? As the MZ B cells are high on IgM this could affect the analysis. Should be gated on follicular B cells.

Yes, Fig 5E is gated on follicular B cells and this has been clarified in the legend.

Referee #3:

This manuscript presents a large amount of data. It describes a research story starting with identifying new components in B-cell receptor (BCR) endocytosis and ending by searching the roles of endophilin A2 in BCR-independent survival and proliferation of peripheral B-cells. This study is the first to use CRISPR-based screening to identify genes regulating BCR-mediated antigen internalization, a critical step in the activation of humoral immune responses. Although the data reveal weaknesses in the cellular assays for the screen, this manuscript presents a new application for the CRISPR-based screening in the field. The screen identified endophilin A2, a component of clathrin-independent endocytosis pathways. The authors developed a human B-cell line, bone marrow chimeric mice, and knockout mice and confirmed the involvement of endophilin A2 in BCR-mediated antigen internalization. However, endophilin A2-dependent BCR internalization did not significantly contribute to BCR-mediated antigen processing and presentation. Instead, endophilin A2 inhibits the turnover and maintains the surface level of the BCR. Therefore, endophilin A2-dependent BCR endocytosis cannot explain the immunological phenotypes of endophilin A2-deficient mice - decreases in follicular and germinal center B cells and IgG levels. After finding that endophilin A2-deficiency inhibits B-cell proliferation in response to a wide arrange of stimuli, the authors turned to examine B-cell metabolism, specifically oxygen consumption rates. The reduction in the oxygen consumption rates leads to the discovery of the defect in iron uptake in endophilin A2-deficient B-cells, including transferrin receptor endocytosis. This study utilized a banquet of cutting edge technologies and generated new information on the cell biology of B-cells, which should be the interest of the readers of EMBO Reports. However, the manuscript lost its focus on endocytosis in the middle and has not got deep into the mechanism by which endophilin A2-mediated endocytosis uniquely regulates B-cell function.

Specific comments:

1. The title is not precise by starting with germinal center B-cell responses. The data presented in the manuscript show that the involvement of endophilin A2 is not exclusive to germinal center B-cells. For example, the number of follicular B-cells was also reduced in endophilin A2-deficient mice. IgG reduction could be due to defects in germinal center B-cells or non-germinal center B-cells, as isotype switching does occur outside germinal centers. The manuscript does not provide any data that compare BCR and transferrin endocytosis in germinal center B-cells with other B-cell subsets. Even though mice were immunized, the majority of their B-cells are not germinal center B-cells.

We have changed the title of the manuscript to reflect the wider defects across B cell populations. However, it is worth noting that in response to antigen GC B cells are prominently affected in a cell-intrinsic manner and thus worth highlighting in the title. We do show that both BCR and TFRC internalization are more profoundly affected in Sh3gl1^{-/-} GC B cells as compared to naïve follicular B cells (Fig 5C, Fig EV7I). Further, in Fig 3F we now show that GC, PC and class-switched IgG⁺ cells are more significantly affected by endophilin A2 deletion than naïve follicular B cells. This is in agreement with our *in vitro* data showing a pronounced growth phenotype of activated B cells that is exacerbated in the presence of competition, as would occur in the GC (Fig EV6E).

2. The BCR endocytosis assay used for screening may not be appropriately performed. This flow cytometry-based BCR endocytosis assay is well established in the literature. The manuscript only shows two small dot plots in Figure 1A. Based on these two plots, the fluorescence of the streptavidin staining (surface BCRs) and the anti-BCR staining (total BCRs) significantly overlap with each other. Such overlap may not ruin the results but dramatically reduced the sensitivity of the screening.

We now show clearer data illustrating the set-up of the screen. In these assays, surrogate antigen typically induced internalisation of about 70% of surface BCR at 30 min in control, non-targeted cells, which is typical for this cell line and time point. In Fig 1, gates indicate sorting of cells that have internalized the majority of the BCR versus those that have not internalized at all. The percent of cells in the internalized gates is thus smaller than what a total quantification of internalisation would be. In addition, a reduction in internalization efficiency in many cells is likely caused by gene-targeting by the library as the percent of cells in the internalized gate is lower than with control cells (shown below). The screen thus should be sensitive to gene effects on the endocytosis. Notably, CRISPR screening in these settings is not particularly sensitive to the exact placement of the gates as it relies on statistics of their comparison, which is mostly affected by library coverage.

3. Endophilin A2 has been shown to contribute to the endocytosis and intracellular trafficking of several receptors in various cell types. After the initial screening with the BCR endocytosis assay, additional assays should be applied to determine if the involvement of endophilin A2 in endocytosis is exclusive to the BCR. Transferrin receptor endocytosis is commonly used for this purpose. If transferrin receptor endocytosis is affected by endophilin A2-deficiency, the endocytosis of additional receptors should be tested to estimate the involvement scope of endophilin A2-dependent endocytosis in B-cells. Under this scenario, the oxygen consumption rate analysis becomes unnecessary.

We agree that this is an interesting point. We think it is likely that endophilin A2 regulates endocytosis of a number of receptors beyond the BCR. We now show in better detail that transferrin receptor internalisation is reduced in endophilin-deficient Ramos and primary B cells. In contrast, internalization of CD40 or BAFFR was normal (data shown below). We now also show that the reduced internalisation of transferrin receptor in endophilin A2-deficient cells causes reduced uptake of transferrin that is more severe than after knock out of EPN1 or PICALM, likely causing the iron deficiency and reduced mitochondrial respiration of endophilin A2-deficient cells. We think that both transferrin receptor endocytosis and mitochondrial respiration are examples of specific cellular defects whose consequences contribute in part to the selective immune phenotype and therefore are important to report in the manuscript. How endophilin A2 regulates transferrin receptor endocytosis remains unclear; our new data show that only a very small fraction of transferrin colocalizes with endophilin A2 spots as compared to clathrin spots, suggesting indirect effects. Similarly, we do not know at the moment why the effects of endophilin A2 deficiency manifest more strongly in GC B cells. However, we feel that addressing these questions is beyond the scope of this study.

4. The antigen internalization versus time plots in Figure 2C and 5A-C were not presented correctly. The time scale is not scientific, which gives a false impression of endocytosis kinetics. If the authors like to link the data points by lines, 5 to 15 min in the x-axis should be three times the length of 0 to 5 min.

Figures 2C and 5A-C have been corrected.

5. Figure 2 should include the percentages of antigen dots colocalizing with endophilin A2 and clathrin using TIRF images. Such data reflect the relative fractions of BCRs that undergo endophilin A2- or clathrin-mediated endocytosis. There are much more of endophilin A2 dots than antigen dots in TIRF images. This brings to the next point: the authors should make sure that the overexpression of fluorescent fusion proteins did not artificially affect B-cells and confirm the critical results by detecting the endogenous endophilin A2, including the pull-down assay in Figure 2H.

We have now quantified the percentage of surrogate antigen spots colocalising with endophilin A2, clathrin or both (Fig 2H). This shows that BCR clusters interact significantly with either CME or FEME sites, but rarely with both simultaneously, strongly suggesting that these endocytic pathways act independently of each other. We have also confirmed the colocalization studies by staining for endogenous endophilin A2 and clathrin in both Ramos cells and primary mouse B cells (data shown below). Unfortunately, the anti-SH3GL1 antibodies available are not suitable for immunoprecipitation and we were thus unable to examine the interaction of the BCR with endogenous endophilin A2 biochemically.

6. Figure 3F should include follicular B-cells.

The proportion of mCherry positive cells in the follicular B cell population is now shown in Fig 3F. These show a decreasing trend over time but do not reach significance. Since these cells do not proliferate in the adoptive transfer model, however, it should be noted that their numbers are very low compared to the GC or plasma cell populations.

7. The effect of endophilin A2-deficiency on the turnover rate of membrane IgM is very interesting, which may be one of the B-cell specific roles of endophilin A2. Does endophilin A2-deficiency have a similar effect on mIgG turnover? The western blots shown in Figure 5F should be labeled with membrane IgM (mIgM) rather than IgM. A molecular weight marker should be added.

Investigating mIgG turnover would be very interesting, unfortunately this is not possible in the primary cells due to the large number of cells required for the assay. An IgG+ immortalised cell line may aid such studies in future. Molecular marker and membrane IgM labels have been added to the Fig 5F.

8. The data transferrin endocytosis in Figure S7G should be included in Figure 6. This is the only data in this manuscript linking endophilin A2-mediated endocytosis to B-cell survival and proliferation.

We have moved the data to Fig 6 and included additional analysis of total transferrin uptake and colocalization of transferrin clusters with clathrin and endophilin A2 (Fig EV7 G, H, I). As noted in the response to comments above, endophilin A2 is required for accumulation of transferrin in cells, however, transferrin clusters at the plasma membrane predominantly associate with clathrin rather than endophilin A2, suggesting endophilin A2 plays an indirect role in iron uptake, possibly through yet unknown effects on intracellular trafficking.

Dear Dr. Tolar

Thank you for the submission of your revised manuscript to EMBO reports. We have now received the reports from referee #1 and #3. While referee 3 is satisfied with the revision and supports publication of the manuscript, referee 1 remains concerned that altered BCR signaling might underlie the reduced IgM levels observed.

You have provided feedback on the remaining concerns and I invite you to address these by adding additional data, as proposed and by a careful discussion of the data and potential alternative interpretations.

From the editorial side, there are also a few things that we need before we can proceed with the official acceptance of your study.

1) Please provide up to 5 keywords.

2) Data availability section:

- Please add a link that resolves to the RNAseq dataset.
- Please deposit the second dataset you have generated, the BioID-based proteomics, in an appropriate public database and list the database and accession code (see <<https://www.embopress.org/page/journal/14693178/authorguide#dataavailability>>).

The format of the Data availability section should follow the model below:

Data availability

3) Datasets:

- Please add the filename (e.g. Dataset EV1) to the legend in the .xls
- Please remove the legends from the Manuscript file.

4) Please note that we can only typeset up to 5 Expanded View figures. Given that some of the EV figures contain only few panels, you could combine some to reduce the number to five. The other alternative is supplying additional figures in the form of an Appendix (single pdf containing figures, legends, a table of content, page numbers).

5) Competing Interests Statement: Please change the header to Conflict of Interest

6) Please correct manuscript section heading

- "Methods" to Materials and Methods.

- and "Extended View legends" to "Expanded View Figure legends"

7) Please change the format of the references to an alphabetical list (Harvard style) instead of a numbered list. You can download the respective EndNote file under this link:
https://endnote.com/style_download/embo-reports/

8) Figure callouts:

- Please add a callout to Fig. EV1D and the panels of Fig. EV4.
- Please change the callouts to the Datasets to 'Dataset EV#' instead of "Supplemental Table #" or "Dataset #".

9) Movies: Please remove the legend from the word manuscript file.

10) Author Checklist Section B: Please include all relevant information on blinding and statistics in a Statistics section in the manuscript.

11) Funding info: We noticed that the same grant number is listed three times from different funders (FC001185). Can you please double-check that this information is correct?

12) Figure legends/Statistics:

- Figure 2D, 2K and EV7I are based on 2 experiments. You already show the individual data points but the error bars and mean value should be removed. The statistical analysis needs to be justified.
- Figure 6G seems to report data from 2 experiments. Please show the individual datapoints instead of the mean with error bars.
- Fig EV5C shows data from 2 mice. Please show the individual datapoints instead of the mean and error bars.
- Figure 4C lacks a definition of sample size.

13) Finally, EMBO reports papers are accompanied online by A) a short (1-2 sentences) summary of the findings and their significance, B) 2-3 bullet points highlighting key results and C) a synopsis image that is 550x200-600 pixels large (width x height) in .png format. You can either show a model or key data in the synopsis image. Please note that the size is rather small and that text needs to be readable at the final size. Please send us this information along with the revised manuscript.

Kind regards,

Referee #1:

Comments to authors

In the revised manuscript, authors majorly do not take into account my suggestions. Particularly, decreased expression of IgM in Fig. 5E and increased expression of Class II in Fig. 4C are very

important in order to explain authors' in vivo data, given the previous accumulating evidence. For example, anergic B cells usually take such low level of surface IgM. However, authors said that we found no significant differences in early BCR signaling, but I guess that authors checked just in vitro signaling capability; probably they did not check careful in vivo BCR signaling readout. So, considering that endophilin A2 is involved in BCR-internalization, I worry about that authors may miss the initial key events to induce such phenotype. In the case of ClassII expression, authors said that this was beyond the current scope.

So, I think that authors still do not reach clarifying the correct underlying mechanisms.

Referee #3:

This is a revised version of a manuscript I have previously reviewed. The revision has addressed most previous comments. However, there are a few minor issues that should be addressed.

1. Fig 2H is not cited in the paper at all.

2. The scale or the unit of the Fig 2H Y-axis looks wrong. If it is percentages, should numbers be timed by 100?

3. Transferrin receptor was abbreviated in the text of the manuscript as TFRC but as TfR in figures. Please be consistent.

RE: EMBOR-2020-51328V2

Dear Martina,

Thank you for the positive progress with our manuscript. We have addressed all the issues that you raised as indicated below. Please find all the changes in the new version of the manuscript and figures

You have provided feedback on the remaining concerns and I invite you to address these by adding additional data, as proposed and by a careful discussion of the data and potential alternative interpretations. Yes, additional BCR signaling data is now included in Fig EV3I and in the text

1) Please provide up to 5 keywords.

Provided in the manuscript

2) Data availability section:

- Please add a link that resolves to the RNAseq dataset.
- Please deposit the second dataset you have generated, the BioID-based proteomics, in an appropriate public database and list the database and accession code (see BioID2 data now deposited in EMBL-EBI BioStudies, Data availability section updated)

3) Datasets:

- Please add the filename (e.g. Dataset EV1) to the legend in the .xls
- Please remove the legends from the Manuscript file.

Complete

4) Please note that we can only typeset up to 5 Expanded View figures. Given that some of the EV figures contain only few panels, you could combine some to reduce the number to five. The other alternative is supplying additional figures in the form of an Appendix (single pdf containing figures, legends, a table of content, page numbers).

Complete, we have combined the previous Fig EV 3, 4 and 5 into one figure and corrected callouts throughout the text

5) Competing Interests Statement: Please change the header to Conflict of Interest

Complete

6) Please correct manuscript section heading

- "Methods" to Materials and Methods.
- and "Extended View legends" to "Expanded View Figure legends"

Complete

7) Please change the format of the references to an alphabetical list (Harvard style) instead of a numbered list. You can download the respective EndNote file under this link:

https://endnote.com/style_download/embo-reports/

Complete

8) Figure callouts:

- Please add a callout to Fig. EV1D and the panels of Fig. EV4.
- Please change the callouts to the Datasets to 'Dataset EV#' instead of "Supplemental Table #" or "Dataset #".

Complete

9) Movies: Please remove the legend from the word manuscript file.

Complete

10) Author Checklist Section B: Please include all relevant information on blinding and statistics in a Statistics section in the manuscript.

Complete

11) Funding info: We noticed that the same grant number is listed three times from different funders (FC001185). Can you please double-check that this information is correct?

Yes, correct as stated

12) Figure legends/Statistics:

- Figure 2D, 2K and EV7I are based on 2 experiments. You already show the individual data points but the error bars and mean value should be removed. The statistical analysis needs to be justified.

- Figure 6G seems to report data from 2 experiments. Please show the individual datapoints instead of the mean with error bars.

- Fig EV5C shows data from 2 mice. Please show the individual datapoints instead of the mean and error bars.

- Figure 4C lacks a definition of sample size.

Complete, apart from 6G - Seahorse data is difficult to present as individual points but each experiment represents several biological replicates

13) Finally, EMBO reports papers are accompanied online by A) a short (1-2 sentences) summary of the findings and their significance, B) 2-3 bullet points highlighting key results and C) a synopsis image that is 550x200-600 pixels large (width x height) in .png format.

Short summary added to the manuscript along with more succinct highlights.

With kind regards,

Pavel Tolar

Dear Martina,

Thank you for the reviewers' comments.

Referee 1 remains concerned that the B cell defect in endophilin A2 knock out mice may be because of altered BCR signalling. Indeed, reduced surface IgM levels and elevated MHC II levels on B cells do sometimes indicate dysregulation of BCR signalling, such as in models of anergy. For this reason, we did analyse BCR signalling in endophilin A2-deficient B cells very rigorously. This included signalling in cell lines and primary murine B cells stimulated in vitro as well as cell activation in primary murine B cells stimulated in vivo. In Figure EV5, we show that endophilin A2-deficient Ramos and primary B cells have normal calcium signalling after BCR triggering. In addition, as shown in the included figure, phosphorylation of key signalling proteins, Syk, Akt and Erk, was similar after BCR triggering in endophilin A2-deficient and wildtype primary cells. We have not included these data in the manuscript to limit the amount of negative data. Critically, we show that the BCR-dependent activation of primary B cells in vivo is not affected by endophilin A2 deficiency, as these cells upregulate normally CD86 and CD69 surface markers after immunisation (Figure 3D and E). All of these readouts are expected to be decreased in anergic cells. Thus, our data strongly argue that endophilin A2-deficient B cells are not anergic and that their BCR signalling supports normal early B cell activation in vivo. This makes it very unlikely that we are "missing" a defect that could better explain the phenotype of the mice. A possible reason is that a small enhancement of BCR signalling caused by the reduced endocytosis triggers an adaptation (e.g. IgM downmodulation), which sufficiently compensates BCR signalling in the endophilin A2-deficient cells. Such a phenotype has a precedent; for example, myosin IIa-deficient B cells show a similar reduction in IgM levels, yet normal BCR signalling (Hoogeboom Cell Reports, 2018). We therefore believe that our data fully address reviewer 1 concerns.

Pavel Tolar
Francis Crick Institute
1 Midland Rd
London, England NW11AT
United Kingdom

Dear Pavel,

Thank you for approving the final minor text corrections. I am now very pleased to accept your manuscript for publication in the next available issue of EMBO reports. Thank you for your contribution to our journal.

At the end of this email I include important information about how to proceed. Please ensure that you take the time to read the information and complete and return the necessary forms to allow us to publish your manuscript as quickly as possible.

As part of the EMBO publication's Transparent Editorial Process, EMBO reports publishes online a Review Process File to accompany accepted manuscripts. As you are aware, this File will be published in conjunction with your paper and will include the referee reports, your point-by-point response and all pertinent correspondence relating to the manuscript.

If you do NOT want this File to be published, please inform the editorial office within 2 days, if you have not done so already, otherwise the File will be published by default [contact: emboreports@embo.org]. If you do opt out, the Review Process File link will point to the following statement: "No Review Process File is available with this article, as the authors have chosen not to make the review process public in this case."

Should you be planning a Press Release on your article, please get in contact with emboreports@wiley.com as early as possible, in order to coordinate publication and release dates.

Thank you again for your contribution to EMBO reports and congratulations on a successful publication. Please consider us again in the future for your most exciting work.

Kind regards,

Martina

THINGS TO DO NOW:

You will receive proofs by e-mail approximately 2-3 weeks after all relevant files have been sent to our Production Office; you should return your corrections within 2 days of receiving the proofs.

Please inform us if there is likely to be any difficulty in reaching you at the above address at that time. Failure to meet our deadlines may result in a delay of publication, or publication without your corrections.

All further communications concerning your paper should quote reference number EMBOR-2020-51328V3 and be addressed to emboreports@wiley.com.

Should you be planning a Press Release on your article, please get in contact with emboreports@wiley.com as early as possible, in order to coordinate publication and release dates.

Corresponding Author Name: Pavel Tolar

Manuscript Number: EMBOR-2020-51328V2